# Influenza A virus surface proteins are organized to help penetrate host mucus

Michael D Vahey[1,2†]*, Daniel A Fletcher[1,3,4]*

[1]Department of Bioengineering, University of California, Berkeley, Berkeley, United States; [2]Biophysics Program, University of California, Berkeley, Berkeley, United States; [3]Biological Systems & Engineering, Lawrence Berkeley National Laboratory, Berkeley, United States; [4]Chan Zuckerberg Biohub, San Francisco, United States

**Abstract** Influenza A virus (IAV) enters cells by binding to sialic acid on the cell surface. To accomplish this while avoiding immobilization by sialic acid in host mucus, viruses rely on a balance between the receptor-binding protein hemagglutinin (HA) and the receptor-cleaving protein neuraminidase (NA). Although genetic aspects of this balance are well-characterized, little is known about how the spatial organization of these proteins in the viral envelope may contribute. Using site-specific fluorescent labeling and super-resolution microscopy, we show that HA and NA are asymmetrically distributed on the surface of filamentous viruses, creating a spatial organization of binding and cleaving activities that causes viruses to step consistently away from their NA-rich pole. This Brownian ratchet-like diffusion produces persistent directional mobility that resolves the virus's conflicting needs to both penetrate mucus and stably attach to the underlying cells, potentially contributing to the prevalence of the filamentous phenotype in clinical isolates of IAV.
DOI: https://doi.org/10.7554/eLife.43764.001

*For correspondence:
mvahey@wustl.edu (MDV);
fletch@berkeley.edu (DAF)

Present address: †Department of Biomedical Engineering, Washington University in St. Louis, St. Louis, United States

Competing interests: The authors declare that no competing interests exist.

## Introduction

For a virus to infect a cell, it must reach receptors on the cell surface while avoiding neutralization or clearance by 'decoy' receptors in the surrounding environment. For viruses that bind to sialic acid, a receptor that is abundant both on the surface of cells and in the secreted extracellular mucosal environment, attachment and detachment is controlled by receptor-binding and receptor-destroying activities on the viral surface (*Rosenthal et al., 1998*; *Zeng et al., 2008*). Influenza A viruses (IAVs), respiratory pathogens that contribute to seasonal flu and have pandemic potential, achieve this balance with hemagglutinin (HA)-mediated receptor binding and neuraminidase (NA)-catalyzed receptor destruction (*Air and Laver, 1989*; *Skehel and Wiley, 2000*). Although the importance and genetic basis of the functional balance between HA and NA in IAV has been thoroughly characterized (*Wagner et al., 2002*; *Xu et al., 2012*; *Yen et al., 2011*), it is becoming clear that transmissibility of influenza viruses within and between hosts depends on factors beyond the sequence of these two genes (*Chou et al., 2011*; *Herfst et al., 2012*; *Neumann and Kawaoka, 2015*).

Mucosal barriers present the first line of defense against IAV infection. To infect the underlying epithelium, viral particles must first pass through a ~ 1–10 μm thick layer of mucus that is being steadily transported towards the pharynx where it can be swallowed, neutralizing any virus immobilized within it (*Bustamante-Marin and Ostrowski, 2017*; *Wanner et al., 1996*; *Zanin et al., 2016*). Viruses that bind too tightly to sialic acid will pass through the mucus barrier very slowly, and will thus be unable to reach the surface of an airway epithelial cell before mucociliary clearance. In contrast, viruses that bind only very weakly to sialic acid (or which rapidly destroy it through excessive NA activity [*Cohen et al., 2013*; *Yang et al., 2014*]) will quickly penetrate mucus, but may be unable to stably attach to the surface of the underlying epithelium once it is reached. Adaptations that help the virus to overcome both of these conflicting challenges, such as changes in virus morphology or a

particular spatial organization of envelope proteins, could be evolutionarily favored during *in vivo* replication.

Interestingly, one feature of IAV that tends to diverge when clinical isolates are cultured in a laboratory environment, or when animals are infected with laboratory-grown strains, is particle morphology. While clinical isolates of IAV – samples adapted to transmission in a mucosal environment – form filamentous particles with a consistent diameter but widely varying length, laboratory-adapted strains tend to produce more uniform, spherical particles (*Badham and Rossman, 2016*; *Chu, 1949*; *Dadonaite et al., 2016*; *Seladi-Schulman et al., 2013*). Recent evidence from the 2009 pandemic suggests that filamentous morphology, conferred by the virus's M segment, may play a role in transmission (*Campbell et al., 2014*; *Lakdawala et al., 2011*). However, whether or not virus morphology contributes directly to virus transmission – and if so, how – remains unclear. Similarly, although the two major envelope proteins of IAV, HA and NA, have been observed by electron microscopy to cluster non-uniformly on both the viral and pre-viral envelope (*Calder et al., 2010*; *Harris et al., 2006*; *Leser and Lamb, 2017*), whether and how the spatial organization of HA and NA affects virus transmission also remains unclear.

Motivated by these observations, we reasoned that virus shape, together with the packaging and organization of HA and NA in the viral membrane, could influence the balance of attachment and detachment in ways that promote efficient virus penetration through mucus. To test this idea, we sought to characterize the organization of proteins in filamentous IAV particles while simultaneously observing their engagement with sialic acid – a measurement that requires a non-destructive approach. To make this measurement possible, we recently developed strains of influenza A virus that are amenable to fluorescence microscopy through site-specific tags introduced into the viral genome (*Vahey and Fletcher, 2019*). Here we show that filamentous particles frequently contain asymmetric distributions of HA and NA in their membranes, and that this distinctive organization biases the diffusion of these particles in a persistent direction over distances of several microns. By enhancing the effective diffusion of a viral particle without reducing the stability of its attachment to the viral receptor, this mechanism could promote virus penetration across mucosal barriers.

## Results

### HA and NA are distributed asymmetrically on the surface of IAV particles

We first sought to characterize the organization and dynamics of proteins in the viral membrane. By labeling HA and NA, along with the viral nucleoprotein, NP, we are able to measure features of virus organization on intact, infectious particles that corroborate and extend previous observations made using electron microscopy (*Calder et al., 2010*; *Chlanda et al., 2015*; *Harris et al., 2006*; *Leser and Lamb, 2017*). For these experiments, we use a tagged variant of the strain A/WSN/1933 with M1 from A/Udorn/1972, which differs from WSN M1 at six residues and confers filamentous morphology (*Elleman and Barclay, 2004*). Consistent with our prior observations, this virus produces filamentous particles that vary widely in size, from sub-diffraction limited spots to particles > 10 µm in length (*Vahey and Fletcher, 2019*). In viruses with fluorescently-labeled HA and NA, we observe a pronounced tendency for NA to be enriched at one end of the virus (*Figure 1A*, inset). To quantify this enrichment, we measured the relative intensities of HA and NA within ~400 nm of each viral pole in filamentous particles > 1 µm in length. Across 20344 such particles, approximately one third have NA intensities at least 5-fold higher on one pole than on the other (*Figure 1A*, left). Similarly, viruses with labeled NP (the most abundant protein in vRNP complexes and a proxy for the virus genome) reveal individual foci that localize to one end of the virus. Aligning filamentous particles with polarized NP foci (4805 in total) reveals a tendency for NA to colocalize with NP (*Figure 1A*, right). For the subset of these particles > 4.5 µm in length (540 in total), we created a composite image showing the average densities of viral proteins within 2 µm of each viral pole (*Figure 1—figure supplement 1*). On average, NA is enriched ~2 x relative to HA at the NP-containing pole in these particles, as compared to the middle region of the particle and the opposite pole (*Figure 1—figure supplement 1*, bottom), suggesting that NA localization is linked (directly or indirectly) to the location of the viral genome. This protein distribution is stable, since photobleached portions of unfixed

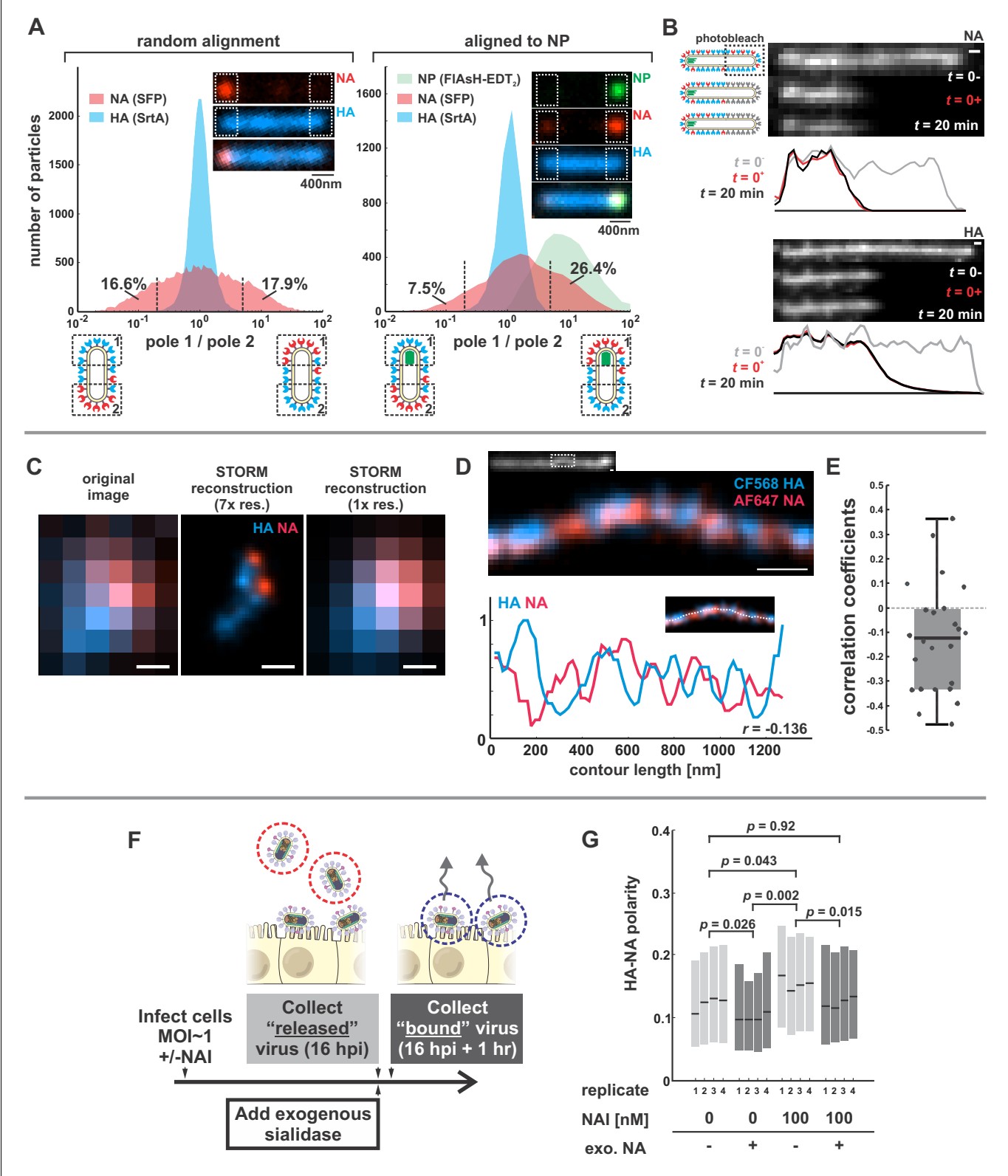

**Figure 1.** Organization of the IAV envelope in filamentous virus particles. (**A**) Abundance of viral proteins at one viral pole relative to the other, measured as the ratio of intensities within ~400 nm of either end of the virus. The plot to the left compares HA and NA intensities for 20344 filamentous particles of at least ~1 μm in length. Approximately one third of filamentous particles have NA intensities five-fold higher on one pole than the other, as indicated by the vertical dashed lines (distributions are symmetric in this case because poles are designated randomly). Inset images show a particle

*Figure 1 continued on next page*

*Figure 1 continued*

with NA enrichment of ~5.6 fold (~31st percentile of particles). The plot to the right shows the same comparison for particles aligned to their NP-foci, designated as pole 1 (distributions are determined from a total of 4805 particles with polarized NP distributions). The prevalence of particles with NA ratios > 1 indicates a tendency for NA to be enriched at the virus's genome-containing pole. Inset images show a particle with NA enrichment of ~3.3 fold (~48th percentile of particles based on NA polarity). (See also *Figure 1—figure supplement 1*).(B) Photobleached fluorescent NA (top) and HA (bottom) on filamentous particles show no recovery after 20 min, indicating that NA and HA are immobilized in the viral membrane (scale bar = 500 nm). Data is representative of *n* = 8 viruses (NA) and *n* = 5 viruses (HA) from three biological replicates. (C) STORM reconstructions at ~30 nm resolution of a pair of viruses unresolveable by diffraction-limited microscopy, each showing the characteristic localization of NA at one end of the virus (scale bar = 200 nm). (D) STORM reconstructions of a filamentous virus at ~30 nm resolution reveal clusters of HA and NA which partly exclude each other, illustrated by the inverse correlation in HA and NA intensities along the axis of the virus (scale bar = 200 nm). (E) Distribution of HA-NA spatial correlation coefficients from regions of 24 filamentous particles following reconstruction at ~30 nm resolution. (F) To compare populations of virus that are able to detach from infected cell to those that remain bound to the cell surface with or without NAIs, we collect virus in two separate stages: first, we harvest virus from media at 16hpi, followed by the addition of media supplemented with an exogenous, oseltamivir-resistant bacterial neuraminidase. After one hour of treatment with exogenous NA, we harvest virus that had previously remained bound to the cell surface. (G) Plot showing the HA-NA polarity of viruses released in the presence or absence of NAI and exogenous sialidase. Boxes show median values of HA-NA polarity and 25th-75th percentile ranges for each of four biological replicates, with between 499 and 2067 filamentous viruses each (*p*-values determined using a paired-sample T-test). See also *Figure 1—source data 1*.

DOI: https://doi.org/10.7554/eLife.43764.002

The following source data and figure supplements are available for figure 1:

**Source data 1.** Matlab source data and code for *Figure 1A and G*.
DOI: https://doi.org/10.7554/eLife.43764.005
**Figure supplement 1.** Distribution of HA, NA, and NP in filamentous particles.
DOI: https://doi.org/10.7554/eLife.43764.003
**Figure supplement 2.** Determining the resolution of virus images reconstructed using STORM.
DOI: https://doi.org/10.7554/eLife.43764.004

filamentous particles do not recover fluorescence in either HA or NA over tens of minutes (*Figure 1B*).

To investigate the finer details of protein organization and to determine if NA in fluorescent viruses is clustered, as suggested by electron microscopy (*Calder et al., 2010*; *Harris et al., 2006*), we use two-color stochastic optical reconstruction microscopy (STORM) to reconstruct images of HA and NA with resolution ~10X better than the diffraction limit (~30 nm compared to ~300 nm; *Figure 1—figure supplement 2*). At this resolution, we find that the tendency of NA to concentrate at one of the viral poles is pronounced even in particles smaller than 300 nm in length (*Figure 1C*). We also find that the NA seen at low levels along the length of filamentous viruses without super-resolution imaging is actually organized into small NA clusters (*Figure 1D*). Additionally, HA and NA distributions in these particles are modestly anti-correlated (*Figure 1E*), suggesting that receptor-binding and receptor-destroying activities may be spatially segregated on some regions of the virus. Collectively, these measurements present a picture of a variegated IAV envelope whose spatial organization is stable and coupled to the presence and location of the viral genome, with ~70% of NP-containing viruses having NA biased to the proximal pole.

We next investigated whether spatial organization of the IAV envelope could have functional significance for virus binding and detachment. As a first test of this idea, we compared the spatial organization of viruses that were released from the cell surface with those that remained attached after challenging the virus with the neuraminidase inhibitor (NAI) oseltamivir carboxylate (*He et al., 1999*) (Materials and methods). To quantify virus spatial organization, we defined HA-NA polarity as the separation between the center of masses for HA and NA divided by the virus length. Interestingly, viruses that escaped the cell surface under NAI challenge showed higher HA-NA polarity (0.155; 95% confidence interval, CI = 0.139–0.171) than viruses released in the absence of NAI challenge, both before (0.123; 95% CI = 0.106–0.140) and after (0.100; 95% CI = 0.090–0.110) facilitating virus release by treating with exogenous sialidase (*Figure 1F & G*). In comparison, viruses treated with both NAI and exogenous sialidase have polarities similar to the untreated virus (0.124; 95% CI = 0.111–0.136). These results suggest that viruses with NA concentrated at one of the viral poles may be more effective at navigating environments rich in sialic acid.

## Polarized viruses step persistently away from their NA-rich pole

To directly test how NA polarization might affect virus motion, we characterized interactions between fluorescently-labeled viruses and sialic acid coated coverslips, where the well-defined geometry and density of sialic acid allow straightforward analysis of virus diffusion (Materials and methods) (*Figure 2A*). Because our approach to fluorescently labeling NA preserves its activity (*Figure 2—figure supplement 1*) and viruses harboring fluorophores on both HA and NA preserve ~85% of their infectivity (*Vahey and Fletcher, 2019*), we could carry out functional assays with the virus and at the same time visualize HA and NA distributions on the viral membrane. Surprisingly, the motion we observed did not resemble randomly-oriented diffusion of the viral particle, but rather persistent, Brownian ratchet-like diffusion, in which filamentous particles with polarized distributions of NA exhibited directed mobility away from their NA-rich pole (*Figure 2B*, *Video 1*). Labeling coverslips with fluorescein-labeled Erythrina cristagalli lectin (ECL), which binds specifically to the terminal galactose exposed following sialic acid cleavage (*Iglesias et al., 1982*), revealed the history of virus trajectories and confirmed that virus motion is accompanied by receptor destruction (*Figure 2B*). Aligning the trajectories of mobile particles (defined as those with diffusion coefficients > 100 nm$^2$/s, comprising ~68% of the population; *Figure 2C*) to the orientation of their HA-NA axis reveals that directional mobility can persist for several microns, many times the length of the particle itself (*Figure 2D*). Consistent with the observation that polarized distributions of HA and NA serve as a determinant for persistent motion and enhanced diffusion, we find that non-mobile viruses have, on average, less polarized distributions of HA and NA than mobile ones (*Figure 2E*).

To determine if polarized distributions of NA were necessary for persistent directional mobility, we disrupted the spatial organization of viral surface proteins by removing the cytoplasmic tail of NA (residues 2–6; *Figure 3—figure supplement 1A*) and rescuing a tagged variant of the virus (NAΔCT). Although NA expression on the surface of infected cells is comparable to that seen for wildtype virus, deletion of the cytoplasmic tail reduces packaging of NA into virions (*Figure 3—figure supplement 1B & C*). Additionally, although NAΔCT virus imaged at ~30 nm resolution using STORM still exhibits NA clusters resembling those found in wildtype virus, the clusters are no longer immobilized on the surface of the virus, as revealed by photobleaching experiments (*Figure 3—figure supplement 2A & B*). Although both wildtype and NAΔCT viruses form filamentous particles (with wildtype virus producing a higher proportion of extremely large filaments >3 μm in length; *Figure 3—figure supplement 2C*), the polarity of NA distributions on the viral surface is significantly decreased in the NAΔCT strain relative to wildtype (*Figure 3—figure supplement 2D*). Using the same sialic acid coated surfaces, NAΔCT viruses no longer exhibited persistent directional motion (*Figure 3A*), indicating that spatial organization of NA at the poles is necessary for persistent mobility of IAV.

Correlations in the direction of successive steps in a random walk will increase the rate at which the walker explores its environment, analogous to how more rigid polymer chains typically have larger end-to-end distances than those with equivalent contour length but lower bending rigidity. To quantitatively compare the rate at which the less persistent NAΔCT viruses explore their environment to that of the more persistent wildtype virus, we determined mean squared displacements from observed trajectories of mobile viruses (diffusion coefficients > 100 nm$^2$/s over the observation period) and simulated random walks in which the number of steps, the size of each step, and the time interval between steps all match the observed data, but the direction of each step is uncorrelated with previous steps (*Figure 3B*). From this analysis, we estimate that the diffusion coefficients of mobile viruses are enhanced approximately five-fold as a result of directional correlations (*Figure 3B*, left). In contrast, the diffusion of mobile NAΔCT viruses is only modestly influenced by directional correlations (~1.5 fold increase; *Figure 3B*, right). To further quantify the difference between viruses with wildtype NA and NAΔCT, we tracked the displacement of particles between subsequent frames acquired 30 s apart and plotted the distribution of the stepping angle relative to the orientation of the virus's HA-NA axis. Viruses with wildtype NA step in the direction of increasing HA (with the NA pole at the rear) roughly twice as frequently as the opposite direction, while NAΔCT viruses exhibit no correlation between orientation and stepping direction (*Figure 3C*). Additionally, despite having lower amounts of NA in the viral membrane, NAΔCT viruses left significantly larger trails of cleaved receptors than their wildtype counterparts (*Figure 3—figure supplement 3*). These

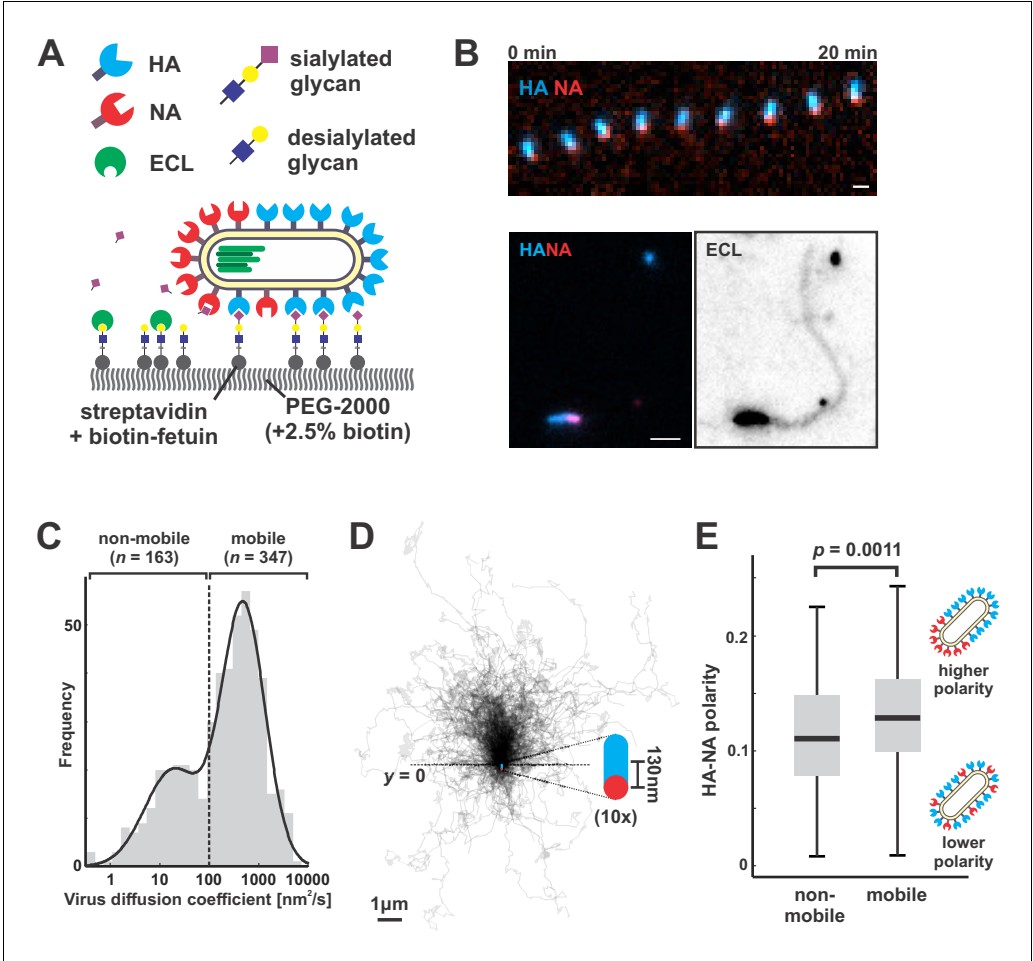

**Figure 2.** Filamentous IAV diffuses via a Brownian ratchet mechanism. (**A**) Labeled viruses are placed on coverslips passivated with PEG2K and functionalized with biotinylated fetuin, which provides a high density of receptors for HA and substrates for NA that can be imaged using TIRF microscopy. (**B**) Time series of a virus migrating in the direction of its higher-HA pole (scale bar = 2 µm). Viruses exhibiting persistent motion on sialic acid-coated surfaces leave a trail of terminal galactose to which fluorescent ECL binds, indicating NA cleavage of receptors as the virus moves. (**C**) Distribution of virus diffusion coefficients determined by measuring the mean squared displacement versus time. Mobile particles are defined as those with diffusion coefficients > 100 nm$^2$/s, corresponding to the more diffusive subset of the bimodally-distributed population. (**D**) Trajectories (measured from timelapse images) of $n$ = 347 mobile viruses registered to their initial positions and aligned based on the orientation of the HA-NA axis. Blue and red dots at $y$ = 0 show the median positions of HA and NA (with median separation of ~130 nm), respectively, across all viruses. Data is pooled from three biological replicates. (**E**) HA-NA organization correlates with virus mobility. Populations exhibiting little motion ('non-mobile') have significantly less polarized distributions of HA and NA than those that exhibit persistent directional mobility (quantification of $n$ = 163 non-mobile and $n$ = 347 mobile viruses combined from three biological replicates; boxes are centered on median values and span from 25th to 75th percentile; $p$-value calculated using a two-sample KS-test). See also *Figure 2—source data 1*.

DOI: https://doi.org/10.7554/eLife.43764.006

The following source data and figure supplement are available for figure 2:

**Source data 1.** Matlab source data and code for *Figures 2C, D and E*.
DOI: https://doi.org/10.7554/eLife.43764.008

**Figure supplement 1.** Effect of fluorescent labeling on NA activity.
DOI: https://doi.org/10.7554/eLife.43764.007

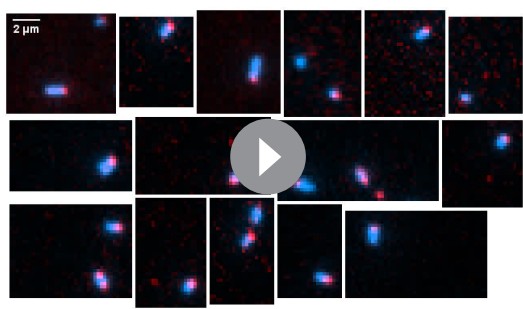

**Video 1.** Montage of IAV particles acquired using total internal reflectance microscopy at 30 s intervals. Labeled HA is shown in blue and labeled NA is shown in red. Panels are shown at equivalent scales.
DOI: https://doi.org/10.7554/eLife.43764.009

results demonstrate that the spatial organization and mobility of NA on the viral surface both play an important role in determining where and when it cleaves substrate.

## Computational modeling suggests a mechanism for persistent directional mobility

To investigate the mechanism of this persistent directional mobility, we modeled the diffusion of filamentous viruses with defined spatial organization of HA and NA as the viruses bind to sialic acid on a two-dimensional surface (Appendix, *Figure 3—figure supplement 4*). In this model, virus diffusion is constrained by the tens of attachments the HAs form with sialic acid at any instant in time (*English and Hammer, 2004*; *Xu and Shaw, 2016*). At the interface between HA-rich and NA-rich regions of the viral membrane, like those seen in fluorescence images (*Figure 1A–D*), NA will periodically gain access to, and hydrolyze, sialic acid as the position of the virus thermally fluctuates. The binding partners available to HAs located close to this interface will therefore be asymmetrically distributed, with more potential binding partners located further away from areas of higher NA density. This concept is illustrated most clearly in the case where the virus is constrained to a one-dimensional path. As this case illustrates, the sialic acid density along the length of the virus is lowest in close proximity to NA (*Figure 3—figure supplement 5A–C*). When NA is confined to a single pole, this results in a gradient in sialic acid density along the length of the particle. Since the number of available binding partners is higher in one direction than another, newly-formed HA-sialic acid bonds will tend to be offset from previous bonds in this direction. This leads to the persistent motion we observe experimentally as well as in our simulations (*Video 2*). Consistent with this mechanism, persistent motion of the virus is lost when NA is no longer localized to a single virus pole and the sialic acid distribution beneath the particle becomes symmetric (*Figure 3—figure supplement 5B–D*). We note that virions with symmetric distributions of NA also occur in the virus population and would not be expected to exhibit persistent motion (for example, those particles with NA polar ratios of ~1 in *Figure 1A*).

Since virus motion is dependent on the virus's ability to establish distinct regions with and without sialic acid, we expect that dynamic distributions of either NA or sialic acid will alter virus mobility. Consistent with our experimental observations with the NAΔCT virus, allowing NA to freely diffuse eliminates directional bias in virus motion in both experiments (*Figure 3C*) and simulations (*Figure 3D*). However, allowing surface-bound sialic acid to diffuse can either enhance or suppress directed motion, depending on the kinetic parameters of HA and NA, the diffusion coefficient of sialic acid ($D_{SA}$), and the size of the virus (Appendix). Although our experimental data corresponds to the case where sialic acid is immobilized ($D_{SA}$ ~0), our simulations predict that polarized IAV bound to slowly-diffusing transmembrane glycoproteins or gel-forming mucins ($D_{SA}$ <0.01 μm²/s) will exhibit enhanced directed mobility, likely due to the biased spatial distributions of sialic acid binding partners that are available to HAs distal to the virus's NA-rich pole. In contrast, increasing the rate of sialic acid diffusion further ($D_{SA}$ >0.01 μm²/s) is expected to suppress persistent motion, by flattening the sialic acid gradient that is created beneath the virus (*Figure 3E*, *Video 3*). Finally, we find that increasing the length of the virus in simulations increases the number of HA-sialic acid interactions, reducing virus mobility but increasing directional persistence (*Figure 3—figure supplement 6*). Thus, the organization of proteins on the viral surface, the size of the virus, and the nature of the receptor to which the virus is bound can each influence the persistence of a virus's motion.

Although our results focus on a two-dimensional geometry in which a virus adheres to a flat surface decorated with sialic acid (a model for the cell surface), we reasoned that they should generalize to three-dimensional systems, such as the secreted mucus gel through which IAV must penetrate to reach naïve cells to infect. To test this prediction, we cultured Calu-3 cells at an air-liquid interface, resulting in a ~ 1–10 μm thick gel of secreted mucus overlaying the apical surface of the cells

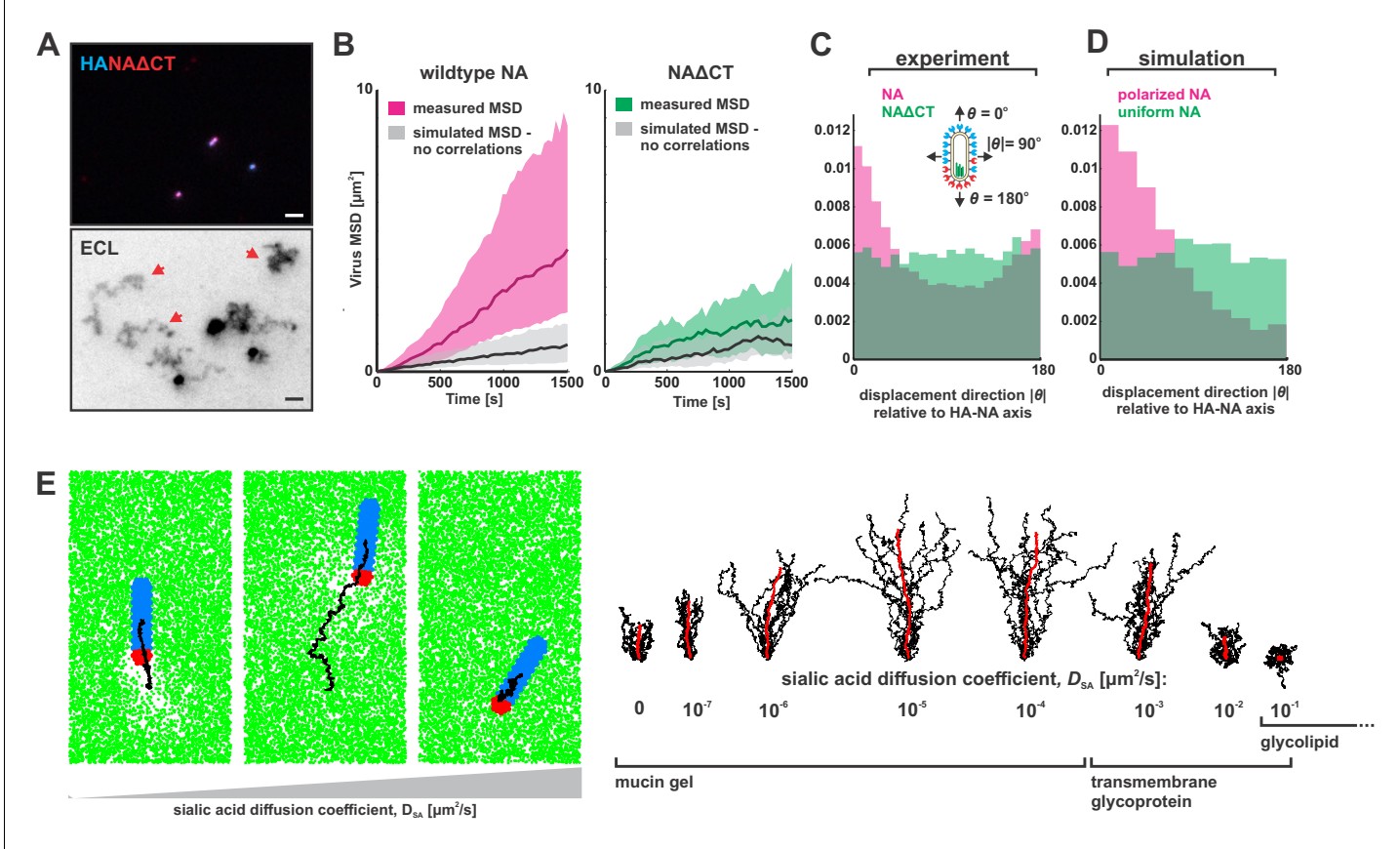

**Figure 3.** Organization of the IAV envelope and diffusion of sialic acid receptors determine persistence of directional mobility. (**A**) NAΔCT virus create ECL tracks that appear less persistent than those generated by virus with wildtype NA. Red arrows indicate tracks where viruses have dissociated (scale bar = 2 μm). (**B**) Mean squared displacements (MSD) for mobile viruses with (left) and without (right) the NA cytoplasmic tail for an observation window of 1500 s. Measured MSDs (in magenta and green) are compared to MSDs from simulated random walks (gray) using the same step size and frequency but with uncorrelated direction. Dark lines show median MSDs, and shaded regions indicate the 25th to 75th percentile range. Data is pooled from three biological replicates. (**C**) Histogram of virus stepping angle relative to virus orientation for virus with wildtype NA (magenta) and NAΔCT (green). While wildtype viruses step in the direction of increasing HA roughly twice as frequently as they step in the opposite direction, virus with NAΔCT do not exhibit any orientational preference. Data is pooled from three biological replicates. (**D**) Quantification for $n = 10$ simulations of 250 nm viruses with uniform (green) or polarized (magenta) distributions of NA, showing the tendency of viruses to step preferentially along the viral axis only when NA is polarized. The model of an idealized virus is in general agreement with experimental results shown in (**C**) that include viruses with a wide variety of sizes and HA-NA organizations. (**E**) The effects of receptor (i.e. sialic acid) diffusion on virus motion. Allowing surface diffusion of sialic acid during simulations can enhance or suppress directional motion of viruses, depending on the receptor mobility and the catalytic rate of NA. Up to a diffusion coefficient of ~0.01 μm²/s, typical of a slowly diffusing membrane protein, a directional bias in virus diffusion is preserved. Trajectories show the results of 10 simulations for each condition in black, with the average is red. See also *Figure 3—source data 1*.

DOI: https://doi.org/10.7554/eLife.43764.010

The following source data and figure supplements are available for figure 3:

**Source data 1.** Matlab source data and code for *Figure 3B, C and D*, *Figure 3—figure supplement 2*, and *Figure 3—figure supplement 3*.
DOI: https://doi.org/10.7554/eLife.43764.017
**Figure supplement 1.** Characterization of influenza A virus lacking the NA cytoplasmic tail (NAΔCT).
DOI: https://doi.org/10.7554/eLife.43764.011
**Figure supplement 2.** Characterization of NAΔCT virus organization and dynamics in the viral membrane.
DOI: https://doi.org/10.7554/eLife.43764.012
**Figure supplement 3.** Comparison of NA activities for soluble and immobilized substrates.
DOI: https://doi.org/10.7554/eLife.43764.013
**Figure supplement 4.** simulation scheme.
DOI: https://doi.org/10.7554/eLife.43764.014
**Figure supplement 5.** Virus organization, sialic acid distributions, and virus trajectories in one- and two-dimensional simulations.
DOI: https://doi.org/10.7554/eLife.43764.015

*Figure 3 continued on next page*

*Figure 3 continued*

**Figure supplement 6.** Contributions of virus morphology and binding affinity to persistent mobility.
DOI: https://doi.org/10.7554/eLife.43764.016

(*Figure 4A*). Adding virus with labeled HA and NA to the mucus, followed by fixation and labeling with ECL, revealed tracks similar to those we observed on two-dimensional surfaces (*Figure 4B*). This suggests that the asymmetry of NA and HA biases the direction of virus diffusion in three dimensions, producing trails of cleaved sialic acid that can reach several microns in length as the virus moves (*Figure 4C*). This directional mobility is less prevalent in viruses with more uniformly distributed NA, though they nonetheless are capable of creating swaths of ECL-staining within the mucus that lack clear directionality (*Figure 4D*). Similar to our observations on idealized two-dimensional surfaces (*Figure 3C*), we find that viruses in mucus exhibit a slight tendency to align their HA-NA axis (captured at the moment of fixation) with the displacement of the virus relative to the center of ECL labeling (*Figure 4E*). These results suggest that the spatial organization of HA and NA on the virus surface may also promote penetration of polarized IAV particles through mucus barriers in vivo.

## Discussion

Advances in electron microscopy over the past several decades have presented an increasingly detailed picture of the morphology and organization of influenza A virus. This has revealed organizational features of filamentous IAV whose origins and functional significance remain unclear. By using site-specific fluorescent labeling to measure the organization of a virus while preserving its function, we are able to corroborate key observations of envelope protein non-uniformity obtained from electron microscopy and extend them to dynamic observations of both protein motion on the surface of the virus, as well as the directionally-persistent motion of influenza A virus particles as they engage with the virus receptor, sialic acid. These observations demonstrate that the morphology of a virus, the spatial organization of proteins in its membrane, and constraints on the diffusion of these proteins collectively confer the tendency for viral particles to exhibit persistent directional mobility, enhancing the virus's rate of diffusion without diminishing its strength of adhesion. Importantly, our simulations show that this feature of filamentous morphology would not be limited to the extremely large particles whose length is easily measured using diffraction limited microscopy, but rather extends to capsule-shaped particles < 200 nm in length (*Figure 3—figure supplement 6*), which are known to be produced by filamentous strains of IAV (*Calder et al., 2010*). In contrast, this effect may be suppressed in spherical particles, due both to the absence of a clear axis for polarization and

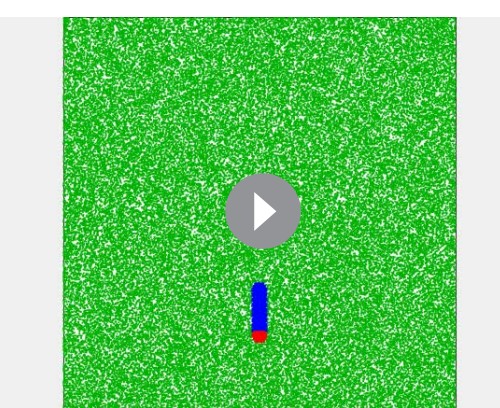

**Video 2.** Simulation of a filamentous virus ~ 250 nm in length with a polarized distribution of HA (blue) and NA (red) on its surface bound to a surface coated with sialic acid (green).
DOI: https://doi.org/10.7554/eLife.43764.018

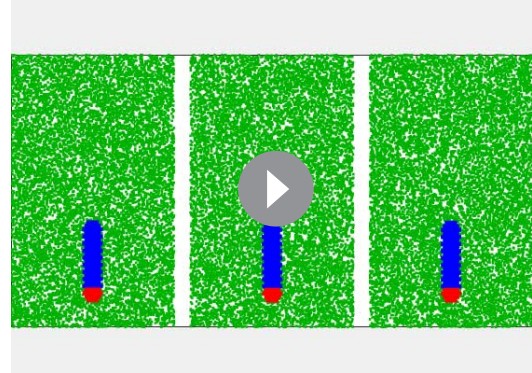

**Video 3.** Simulations of polarized filamentous viruses on surfaces with freely-diffusing sialic acid (green). Simulations correspond to $D_{SA} = 10^{-7}$ μm²/s, $D_{SA} = 10^{-5}$ μm²/s, and $D_{SA} = 10^{-1}$ μm²/s, as plotted in *Figure 3E*.
DOI: https://doi.org/10.7554/eLife.43764.019

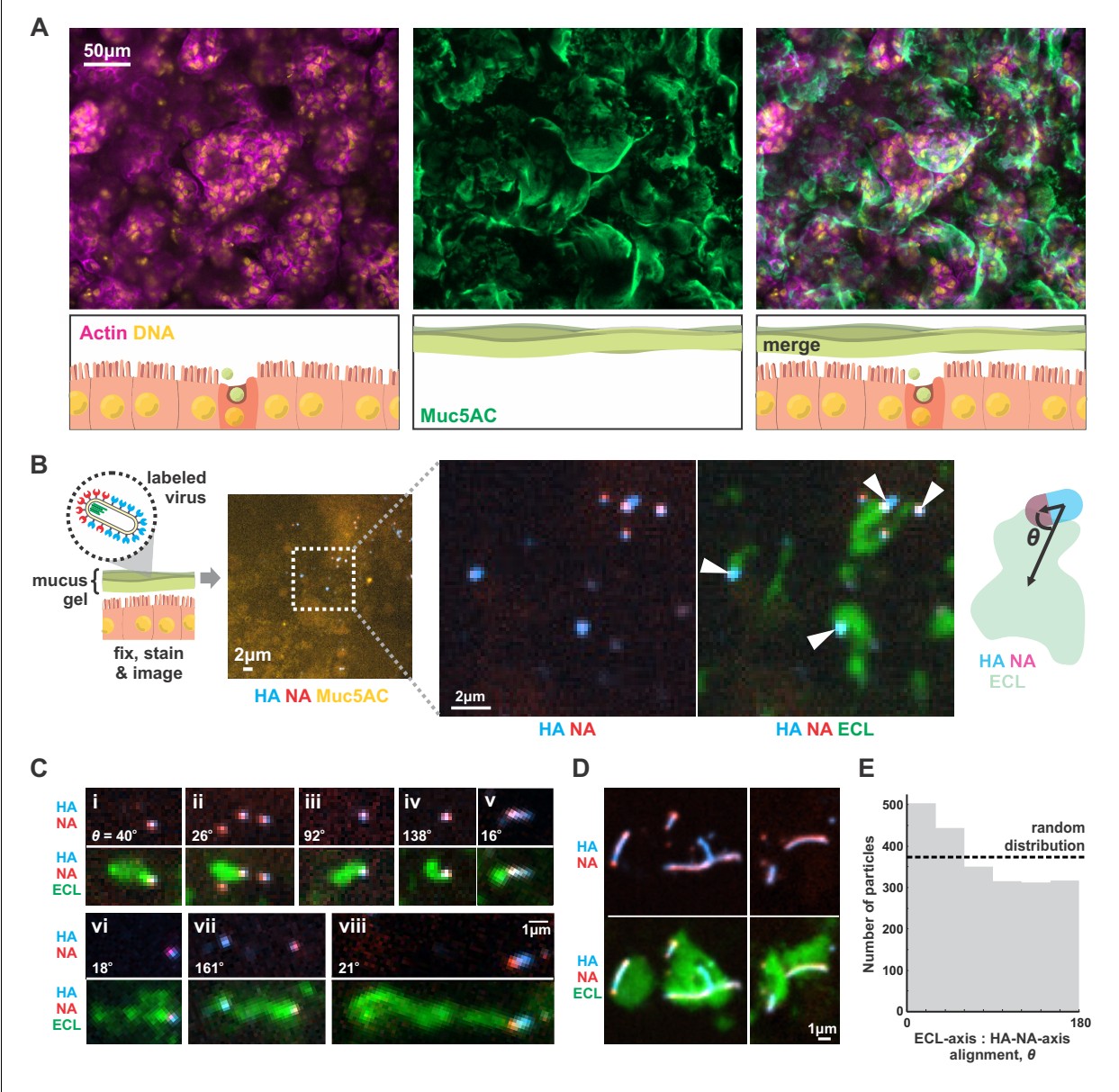

**Figure 4.** IAV exhibits persistent directional mobility in native mucus gels. (**A**) Calu-3 cells cultured at an air-liquid interface for ~10 days partially differentiate, secreting gel-forming mucins (visualized with an antibody against Muc5AC). (**B**) Labeled virus (fluorescent HA and NA) added to the mucosal surface bind and diffuse. Labeling with ECL reveals distinct tracks trailing up to several microns behind viruses (indicated with white arrows). The angle θ characterizes the alignment between the HA-NA axis of the virus and the displacement from the centroid of ECL labeling. (**C**) Panels (i-viii) of registered viruses showing ECL trails. Where polarized distributions of NA on the virus are visible (i, v, vi, viii), ECL labeling is most prominent trailing from the NA-rich pole. Calculated values of the angle θ are given in the lower left of each panel. (**D**) Larger filamentous virions with NA broadly distributed across their surfaces clear large patches of sialic acid on the mucosal surface. Images in B, C, and D are a representative sampling from three biological replicates. (**E**) Orientation of virus HA-NA axes with ECL displacement vectors (the angle θ defined schematically in B). The distribution shows alignments for 2242 virus particles with non-overlapping ECL tracks, pooled from three biological replicates. Dashed line indicates the expected level for a uniform random distribution of alignments. See also **Figure 4—source data 1**.

DOI: https://doi.org/10.7554/eLife.43764.020

The following source data and figure supplement are available for figure 4:

**Source data 1.** Matlab source data and code for **Figure 4E**.

DOI: https://doi.org/10.7554/eLife.43764.022

**Figure supplement 1.** A first passage model for virus transport in mucus.

DOI: https://doi.org/10.7554/eLife.43764.021

their ability to roll on two-dimensional surfaces. However, super-resolution live imaging will be required to confirm these predictions experimentally.

Although more work is needed to understand if these characteristics of IAV organization enhance infectivity and are adaptive during in vivo replication, a simple model of viral transport suggests how the directional viral mobility we observe could contribute during host-to-host transmission. On first entering the respiratory tract, viruses must diffuse through the mucosal barrier to infect the underlying airway epithelial cells. Competing with this process is mucocilliary clearance, which carries particles bound to mucus towards the pharynx where they are neutralized (*Figure 4—figure supplement 1A*). If the transport of a virus through a mucosal barrier is modeled as a first passage problem with a time limit imposed by the rate of mucociliary clearance, a five-fold increase in the diffusion coefficient (as predicted in *Figure 3B*) would lead to a proportional reduction in the first passage time. More dramatically, it would also increase the fraction of particles that reach the cell surface when the rate of mucociliary clearance is high and would normally prevent most particles from reaching the cell surface. Our experiments on two-dimensional sialic acid-coated surfaces suggest a diffusion coefficient of ~800 $nm^2$/s for polarized viruses; adapting this value for a one-dimensional first passage model suggests that persistent motion increases the number of particles that reach the epithelium before being cleared by several orders of magnitude (Appendix, *Figure 4—figure supplement 1B*). Although there are multiple ways that a virus's passage through mucus can be accelerated (reducing its size, decreasing the number of HAs on its surface, increasing the number of NAs, or making the spatial distribution of NA more uniform), each of these changes would likely reduce binding stability once the virus has managed to reach the cell surface. In contrast, the polarized distributions of NA on the viral surface that we observe increase virus diffusion without compromising binding stability.

Balancing the need to robustly attach to the cell surface with the need to escape immobilization in host mucus is a fundamental challenge for influenza and other viruses that use sialic acid to enter cells. Previous work has shown that influenza particles are not immobilized during their interactions with sialic acid (*Guo et al., 2018*; *Sakai et al., 2017*). Additionally, work characterizing the diffusion of influenza C virus, in which receptor-binding and receptor-destroying activities are combined in a single protein (HEF), has shown that the destruction of receptors on a 2D surface prevents virions from retracing their steps (*Sakai et al., 2018*). Our results show that influenza A virus accomplishes a similar feat through a different mechanism – asymmetric distribution of receptor-binding and receptor-destroying activities on the viral surface that results in persistent motion while maintaining stable attachment. The persistent directional mobility shown here orients filamentous virus motion parallel to its major axis, where it could further enhance the anomalous diffusion coefficients observed for cylindrical nanoparticles in mucus (*Yu et al., 2016*). Analogous to the combined importance of active motility and cell shape in mucosal bacteria (*Bartlett et al., 2017*; *Sycuro et al., 2010*), the directed mobility and cylindrical shape of IAV could help to explain the virus's ability to penetrate host mucus and contribute to the prevalence of filamentous morphology in clinical isolates of IAV.

# Materials and methods

## Key resources table

| Reagent type (species) or resource | Designation | Source or reference | Identifiers | Additional information |
|---|---|---|---|---|
| Cell line (*C. familiaris*) | MDCK-II | UC Berkeley Cell Culture Facility | | https://bds.berkeley.edu/facilities/cell-culture |
| Cell line (*H. sapiens*) | Calu-3 | UC Berkeley Cell Culture Facility | | https://bds.berkeley.edu/facilities/cell-culture |
| Cell line (*H. sapiens*) | HEK293T | UC Berkeley Cell Culture Facility | | https://bds.berkeley.edu/facilities/cell-culture |
| Antibody | anti-Muc5AC | Thermo Fisher | RRID:AB_10978001, Cat. #: MA5-12178 | IF (1:400) |

*Continued on next page*

*Continued*

| Reagent type (species) or resource | Designation | Source or reference | Identifiers | Additional information |
|---|---|---|---|---|
| Antibody (lectin) | FITC-labeled Erythrina cristagalli lectin (ECL) | Vector | RRID:AB_2336437, Cat. #: FL-1141 | IF (1:1000) |
| Recombinant DNA reagent | pcDNA3.1+vHA(G5) | (*Vahey and Fletcher, 2019*) | | |
| Recombinant DNA reagent | pcDNA3.1+vHA | (*Vahey and Fletcher, 2019*) | | |
| Recombinant DNA reagent | pcDNA3.1+vNA(ybbR) | (*Vahey and Fletcher, 2019*) | | |
| Recombinant DNA reagent | pcDNA3.1+vNAΔCT(ybbR) | This paper | | Vector shown schematically in *Figure 3—figure supplement 1A* |
| Recombinant DNA reagent | pcDNA3.1+vNA | (*Vahey and Fletcher, 2019*) | | |
| Recombinant DNA reagent | pcDNA3.1+vNP(FlAsH) | (*Vahey and Fletcher, 2019*) | | |
| Recombinant DNA reagent | pcDNA3.1+vNP | (*Vahey and Fletcher, 2019*) | | |
| Recombinant DNA reagent | pcDNA3.1+vM (Ud M1) | (*Vahey and Fletcher, 2019*) | | |
| Recombinant DNA reagent | pcDNA3.1+vN | (*Vahey and Fletcher, 2019*) | | |
| Recombinant DNA reagent | pcDNA3.1+vPA | (*Vahey and Fletcher, 2019*) | | |
| Recombinant DNA reagent | pcDNA3.1+vPB1 | (*Vahey and Fletcher, 2019*) | | |
| Recombinant DNA reagent | pcDNA3.1+vPB2 | (*Vahey and Fletcher, 2019*) | | |
| Peptide, recombinant protein | Streptavidin | Thermo Fisher | Cat. #: 434302 | |
| Peptide, recombinant protein | Neuraminidase from Clostridium perfringens | Sigma | Cat. #: N2876 | |
| Peptide, recombinant protein | Fetuin from fetal bovine serum | Sigma | Cat. #: F3004 | |
| Peptide, recombinant protein | CLPETGG peptide | Genscript | | |
| Chemical compound, drug | Alexa Fluor 555 maleimide | Thermo Fisher | Cat. #: A20346 | |
| Chemical compound, drug | Alexa Fluor 647 maleimide | Thermo Fisher | Cat. #: A20347 | |
| Chemical compound, drug | CF568 maleimide | Biotium | Cat. #: 92024 | |
| Chemical compound, drug | FlAsH-EDT2 | Toronto Research Chemicals | Cat. #: F335200 | |
| Software | Matlab | Mathworks | | |

## Culturing and labeling virus

Strains of influenza A virus amenable to site specific labeling on HA, NA, and NP were designed and characterized as described previously (*Vahey and Fletcher, 2019*). Briefly, viruses expressing HA with five consecutive glycine residues following the signal sequence (for labeling via Sortase A [*Theile et al., 2013*]), NA with a c-terminal ybbR tag (for labeling via Sfp [*Yin et al., 2006*]), and NP with a c-terminal tetracysteine motif (for labeling via direct binding of the biarsenical dye FlAsH [*Griffin et al., 1998*]) were rescued using reverse genetics (*Hoffmann et al., 2000*) by transfecting

co-cultures of HEK293T and MDCK-II cells with plasmids encoding each of the eight genomic segments under the control of bidirectional promoters. Cells used in this work were obtained and authenticated by the UC Berkeley Cell Culture Facility and tested negative for mycoplasma. All viruses used in this work are derived from A/WSN/1933, with the WSN M1 gene replaced by that of A/Udorn/1972, to establish the filamentous phenotype. MDCK-II cells used to propagate virus were maintained in DMEM supplemented with 10% fetal bovine serum (Thermo Fisher, 10438026) and 1x penicillin/streptomycin (Thermo Fisher, 15140122). Prior to infection, confluent monolayers of cells were washed once with PBS, and serum-containing growth media was replaced with virus growth media (MEM, 0.25% BSA, 1 µg/ml TPCK-treated trypsin, and penicillin/streptomycin). Viruses used for experiments were collected from cells following infection at MOI ~ 1 and 16 hr of growth at 37°C in virus growth media. Media containing virus was collected, centrifuged at 2000 × g for five minutes to remove cell debris, and treated with 10mU/ml soluble sialidase (from *C. Perfringens*, Sigma N2876) to ensure that viruses were well dispersed. Viruses were labeled in solution for 90 min at room temperature using NTC buffer (100 nM NaCl, 20 mM Tris pH 7.6, 5 mM CaCl$_2$) supplemented with 5 mM MgCl$_2$, Sortase A (180 µM enzyme, 50 µM CLPETGG peptide) and SFP synthase (5 µM enzyme, 5 µM CoA probe). Following labeling, Capto Core 700 beads (GE Healthcare;~1:1 resin volume to sample volume) were used to remove residual dyes and enzymes from the solution of labeled virus. For labeling NP with the biarsenical dye FlAsH, viruses were immobilized on coverslips, washed in NTC buffer, and incubated with 2 µM FlAsH for 30 min at room temperature.

## Virus photobleaching

To qualitatively evaluate the mobility of HA and NA on the virion surface (*Figure 1B*, *Figure 3—figure supplement 2B*), unfixed, immobilized viruses were imaged using total internal reflectance (TIRF) microscopy. Filamentous virus of sufficient length (>5 µm) were positioned within the field of view such that when the field stop was closed, only approximately half of the virus was exposed to illumination. This half of the virus was then bleached using maximum laser power and then imaged at lower power at 30 s intervals as the field stop was opened to monitor recovery. Representative results are shown in *Figure 1B* and *Figure 3—figure supplement 2B*.

## STORM imaging and analysis

Samples for STORM imaging were prepared by binding labeled virus to antibody or sialic acid (i.e. fetuin) functionalized coverslips for one hour at 4°C, followed by the incubation of Dragon Green-labeled 220nm-diameter streptavidin coated beads. Viruses and beads were then fixed to the surfaces with 4% paraformaldehyde in PBS and washed 3x with buffer containing 1M Tris pH 8.0, 5% glucose, and 140 mM β-mercaptoethanol. Following these washes, the buffer was supplemented with glucose oxidase and catalase to final concentrations of 0.6 mg/ml and 0.035 mg/ml, respectively, and mounted on the microscope for imaging.

STORM data was acquired in the following sequence. First, an image of the Dragon Green beads (serving as fiducial marks), HA (labeled with CF568-conjugated peptide and SrtA), and NA (labeled with AF647-CoA and SFP) was acquired, to enable registration. Next, a sequence of STORM images was acquired using a 640 nm laser at full power, with acquisition of the HA channel via a 560 nm laser at low power every 50 frames to correct for drift. After collecting 15000–35000 frames in this way, we performed STORM imaging on CF568-HA using a 560 nm laser. To correct for drift, we acquire an image every 50 frames using a 405 nm laser and 575/20 nm emission filter; these settings allow us to image the Dragon Green beads, while simultaneously accelerating blinking of the CF568 dye. For reconstructions of HA, we acquire 25000–35000 frames.

For quantification and localization of blinking events, we use the ImageJ plugin Thunderstorm (*Ovesný et al., 2014*), combined with custom Matlab scripts for additional drift correction and removal of fluorophores that remain in the 'on' state for more than one frame. This analysis results in a list of coordinates for each localization that we then use to reconstruct images of virus at varying resolutions. Reconstructed STORM images (e.g. *Figure 1C & D*) are displayed by representing each localization as a gaussian with a standard deviation of 30 nm.

## NAI challenge assay

Challenge experiments with the neuraminidase inhibitor oseltamivir are performed as described previously (*Vahey and Fletcher, 2019*). The analysis from *Figure 1E* uses an image dataset from *Vahey and Fletcher (2019)*, reanalyzed to measure the spatial organization of HA and NA on the surface of released virus particles. We infect a polarized monolayer of MDCK cells grown on a collagen gel at MOI of 1–3. After incubating cells with virus for one hour at 37°C, we wash to remove excess virus, replacing media with virus growth media supplemented with or without a specified concentration of oseltamivir carboxylate (Toronto Research Chemicals O700980), but without TPCK-treated trypsin. At 16 hr post infection, we remove the virus containing media for labeling and imaging, and replace with media supplemented with 1 U/ml NanI from *C. perfringens* (Sigma N2876). After treating with this exogenous sialidase for one hour at 37°C, we again collect cell culture media for virus labeling and imaging.

To measure the HA-NA polarization on viruses released in these experiments, we segment filamentous viruses with lengths > 1 μm and measure the intensity-weighted centroid (i.e. 'center of mass') for both the HA and NA channels. The vector connecting the NA centroid to the HA centroid defines the orientation of HA-NA polarity. Dividing the magnitude of this vector by the total particle length gives the HA-NA polarity metric plotted throughout this work.

## Virus motility assay

Coverslips presenting sialic acid for virus attachment were prepared as described previously (*Vahey and Fletcher, 2019*). Briefly, NH2-PEG-OH (Rapp Polymere, 122000–2) supplemented with 2.5 mole-percent NH2-PEG-Biotin (Rapp Polymere, 133000-25-20) was conjugated to silanized coverslips for subsequent attachment of sialylated proteins. Following PEGylation, custom PDMS chambers were attached to coverslips, and chambers were incubated for 10 min at room temperature with streptavidin at 50 μg/ml in 150 mM NaCl, 25 mM HEPEs, pH 7.2, and washed 5X in the same buffer. Fetuin (Sigma F3004) labeled with NHS-biotin was then added at 100 nM and incubated ~30 min at room temperature. Coverslips were then washed 5X in NTC buffer and equilibrated to 4°C in preparation for virus binding. For surfaces functionalized in this way (*Piehler et al., 2000*), we expect a PEG density of ~0.75 molecules/nm$^2$, corresponding to roughly one PEG-Biotin per 7 nm x 7 nm area on the surface. If each biotin is bound by one streptavidin tetramer and subsequently one molecule of biotinylated fetuin (with ~10 sialic acid residues per molecule), the surface density of sialic acid will be ~0.2 SA/nm$^2$.

Viruses with HA and NA enzymatically labeled as described previously were bound to coverslips equilibrated to 4°C for one hour on ice. Immediately before imaging, excess virus was washed with pre-chilled NTC, and the sample was mounted on the microscope stage. After allowing the chamber to equilibrate to room temperature, the buffer in the chamber was exchanged to NA buffer (100 mM NaCl, 50 mM MES pH 6.5, 5 mM CaCl$_2$), and timelapse recordings were collected using TIRF microscopy at 30 s intervals. To visualize trails of cleaved sialic acid, fluorescein-labeled *Erythrina cristagalli* lectin (ECL; Vector Labs FL-1141) was added to each well at a concentration of 5 μg/ml in NTC buffer with 10 mg/ml BSA and incubated for 30 min at room temperature. During this incubation period, rapid multivalent binding of ECL to cleaved sialic acid on the viral surface effectively blocks further motion of the virus. Images of ECL trails were acquired using TIRF microscopy without washing unbound ECL from the chamber.

To analyze images of viruses, we separately segment each image according to the intensity in both the HA and NA channels. Merging the two sets of segmented images allows us to determine the position of each virus (i.e. the centroid of the combined HA and NA masks) as well as their morphological features. To determine the HA-NA polarization of each virus, we calculate the distance between the centroid of HA and NA intensity for a particular virus, divided by that virus's length. The data in *Figure 2* and *Figure 3* is compiled by extracting these features for each virus in each frame of a timelapse recording.

To analyze the trails of cleaved sialic acid left by mobile viruses, we segment images of samples labeled with fluorescent ECL according to the intensity of lectin staining. Because the virus itself contains high densities of sialic acid on its surface, we use a bandpass threshold to specifically quantify ECL bound to processed glycans on the coverslip (which produces a signal brighter than the background, but dimmer than the virus). Although viruses dissociate from some of the tracks, those that

remain bound to the surface allow us to connect intensity of ECL labeling on the surface to intensities of HA and NA on the virus that generated the track. This data is plotted in *Figure 3—figure supplement 3*, with and without normalization to NA intensity.

## Quantification of NA activity using MUNANA

For assays using the fluorogenic neuraminidase substrate MUNANA, 10 µl of solution containing labeled virus was diluted into 40 µl of NA buffer (100 mM NaCl, 50 mM MES pH 6.5, 5 mM CaCl$_2$) and incubated in a test tube at 37°C. At time points of 0, 30, 60, 120, and 180 min, aliquots were collected and NA was inactivated by adding sodium carbonate to a final concentration of 100 mM, and fluorescence was measured by imaging a fixed volume of sample on a confocal microscope using excitation at 405 nm. The rate of turnover was then determined from the slope of the intensity versus time plot with the signal at 0 min subtracted from each timepoint. Because strains with wildtype NA and NAΔCT produce different titers of virus that also differ in their morphology and NA composition, we normalized samples to total NA content (*Figure 3—figure supplement 3A*), measured by immobilizing viruses on coverslips, imaging them, and integrating the total NA intensity associated with the two samples. This yields an estimated 6-fold difference in total NA content between virus with wildtype NA and viruses with NAΔCT.

## Virus imaging at air-liquid interface of Calu-3 cell cultures

Calu-3 cells grown on plastic dishes for fewer than 10 passages (DMEM, 10% FBS, 1x penicillin/ streptomycin, supplemented with 1 mM sodium pyruvate (Thermo Fisher, 11360070) and 1x non-essential amino acids (Thermo Fisher, 11140076)) were split at 80% confluence and seeded onto 6 mm transwell supports at 50000 cells per insert. Approximately 3 days after seeding, media from the apical compartment was removed (the 'airlift') and cells were provided with fresh media in the basal compartment every other day until being collected for experiments, 8–12 days following the airlift. Cells cultured in this way differentiated into a secretory phenotype, producing a layer of mucus ~1– 10 µm thick over the apical surface of the monolayer (*Figure 4A*). To bind virus without washing away secreted mucus, 5–10 µl of labeled virus was added to the apical side of each transwell insert (enough to just coat the surface), and immediately removed, leaving ~1 µl of residual virus-containing solution that is approximately evenly distributed on the surface of the monolayer. The cells were then returned to the incubator for 3–6 hr, allowing virus to bind and diffuse, and allowing some of the residual moisture to dry before the cells are collected and fixed on ice for 20 min using 4% paraformaldehyde in PBS supplemented with 1 mM CaCl$_2$. Following fixation, cells and mucus were labeled with MUC5AC monoclonal antibody (45M1; MA5-12178 ThermoFisher) and Erythrina crista-galli lectin labeled with FITC (5 µg/ml; Vector Laboratories, FL-1141). Following labeling, the transwell insert was carefully excised with a razor blade and inverted onto a coverslip for imaging.

To quantify alignment between the HA-NA axis of a virus particle and the direction of its displacement from the associated ECL track (*Figure 4E*), fluorescent images of HA, NA, and ECL were segmented in both HA (to identify virus particles) and ECL (to identify regions of cleaved sialic acid) channels. For each segmented virus particle, we calculate the HA-NA axis (defined as the vector displacement between the center of masses for HA and NA) and the ECL displacement (defined as the vector displacement between the center of masses for HA and ECL) (*Figure 4B*, right). Cases where multiple particles contact the same ECL track are excluded from analysis.

## Statistics and replicates

Replicates referenced throughout this paper refer to biological replicates, defined as virus collected from separate infected cell cultures, labeled, and assayed as indicated. No statistical methods were used to predetermine sample size. Image data was excluded from analysis in rare cases if the sample drifted on the microscope stage, or if coverslip preparations showed non-specific virus binding. Statistical tests and the number of replicates used in specific cases are described in figure captions. All statistical tests were performed in Matlab R2017b using the Statistics and Machine Learning Toolbox. Confidence intervals for the data in *Figure 1G* were calculated using a critical value of the Student's t distribution of 3.182 (95% confidence interval for *n*-1 = 3 degrees of freedom).

## Acknowledgements

The authors would like to acknowledge members of the Fletcher Lab for feedback and technical consultation. This work was supported by the NIH R01 GM114671 (DAF), the Immunotherapeutics and Vaccine Research Initiative at UC Berkeley (DAF), and the Chan Zuckerberg Biohub (DAF). MDV was funded by a Burroughs Wellcome Fund CASI Fellowship. DAF is a Chan Zuckerberg Biohub investigator.

## Additional information

### Funding

| Funder | Grant reference number | Author |
| --- | --- | --- |
| National Institutes of Health | GM114671 | Daniel A Fletcher |
| University of California Berkeley | Immunotherapeutics and Vaccine Research Initiative | Daniel A Fletcher |
| Burroughs Wellcome Fund | CASI Award 1013923 | Michael D Vahey |
| Chan Zuckerberg Biohub | | Daniel A Fletcher |

The funders had no role in study design, data collection and interpretation, or the decision to submit the work for publication.

### Author contributions

Michael D Vahey, Daniel A Fletcher, Conceptualization, Resources, Data curation, Formal analysis, Supervision, Funding acquisition, Investigation, Visualization, Methodology, Writing—original draft, Project administration, Writing—review and editing

### Author ORCIDs

Michael D Vahey (ID) http://orcid.org/0000-0001-9453-4860
Daniel A Fletcher (ID) http://orcid.org/0000-0002-1890-5364

### Decision letter and Author response

Decision letter https://doi.org/10.7554/eLife.43764.027
Author response https://doi.org/10.7554/eLife.43764.028

## Additional files

### Supplementary files

• Source code 1. Matlab code for modeling the diffusion of virus particles bound to sialic acid on two-dimensional surfaces. The folder 'Immobile SA' simulates the case where sialic acid is stationary on the surface, while the folder 'Mobile SA' simulates the case where sialic acid is able to freely diffuse (unless bound by the virus). The function run_simulations.m can be used to launch simulations in both folders. More information is contained in the documentation of each function.
DOI: https://doi.org/10.7554/eLife.43764.023

• Transparent reporting form
DOI: https://doi.org/10.7554/eLife.43764.024

### Data availability

All data generated or analysed during this study are included in the manuscript and supporting files.

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

## Appendix 1

DOI: https://doi.org/10.7554/eLife.43764.025

## Simulations of virus binding and diffusion

### Goals of the model

We wanted to develop simulations that would provide insight into the mechanisms of the directional mobility of influenza A virus. Because individual viruses are highly variable (differing in shape, composition, and organization) and the dynamics of a cylindrical particle close to a surface are complex, we did not seek to develop a model that quantitatively predicts the rates of virus migration that we observe experimentally. Instead, we focused on predicting the general phenomenon of organization-dependent directional mobility, and how this mobility changes when characteristics of the virus or its receptors (i.e. sialic acid, SA) are altered.

### Modeling HA-SA binding and NA-SA cleavage

We model sialic acid coated surfaces as a uniform random distribution of SA at a density of 0.02 /nm$^2$. Although this value is lower than the estimated density in our experiments as well as on the surface of a cell (~0.1–1 SA/nm$^2$) (*Rosenberg and Einstein, 1972*), it serves as a reasonable approximation given that HAs do not bind all SA glycoforms with similar affinity, and also that not all SAs on the surface of fetuin may be accessible for binding after the protein is immobilized on the surface. For HA and NA distributions on the surface of the virus, we use a triangular lattice with a spacing of 15 nm between neighboring spikes, with 84 HAs and 16 NAs proximal to the surface for a filamentous virus 300 nm long. Although we do not explicitly account for the trimeric/tetrameric nature of HA/NA, this should have relatively little effect on our results since each HA can bind more than one SA and each NA can cleave multiple SAs. Except in simulations where NA is allowed to diffuse on the viral surface (modeling the behavior of NAΔCT that we observe experimentally, *Figure 3—figure supplement 2B*), the locations of HA and NA on simulated viruses are static, translating and rotating with the virus as a rigid object. Although this is consistent with our measurements of protein mobility on the viral surface (*Figure 1B*), it does not account for rotation of the virus around its minor axis, which could occur in our experiments.

At each time step in a simulation, we identify HA-SA and NA-SA pairs that are within a radius $b$ of each other; we define these SA as being available for either binding (if close to HA) or cleavage (if close to NA). The radius $b$, which we take to be 7.5 nm, models the flexibility of the carbohydrate and protein backbone to which SA is attached, along with the potential flexibility of HA and NA on the surface of the virus. For HA binding kinetics, we used $k_{on}$ = 400–1000 M$^{-1}$s$^{-1}$ and $k_{off}$ = 1 s$^{-1}$, based on previous measurements (*Sieben et al., 2012*; *Takemoto et al., 1996*). The probabilities of a particular HA-SA pair within a radius $b$ of each other forming ($p_{on}$) or losing ($p_{off}$) a bond in a time interval $\Delta t$ are therefore approximated by:

$$p_{on} \approx \frac{3k_{on}\Delta t}{\frac{4}{3}\pi 1000 b^3 N_{Av}} \approx \frac{k_{on}\Delta t}{1000 b^3 N_{Av}} \tag{1a}$$

$$p_{off} \approx k_{off}\Delta t \tag{1b}$$

The factor of 1000 in the denominator of the expression for $p_{on}$ converts the volume accessible to an HA molecule to units of liters. Since the concentration of sialic acid within a binding radius is high relative to the $K_m$ for NA (>10 mM versus 100 μM-1mM [*Benton et al., 2015*]), the probability of an NA-SA pair resulting in cleavage is determined from the catalytic rate constant ($k_{cat}$ ~100 s$^{-1}$) as:

$$p_{cut} \approx k_{cat}\Delta t \tag{2}$$

For the typical time steps used in our simulations (~0.2 s), this amounts to instantaneous cleavage of any SA residues within reach of NA. Using these three probabilities, we calculate at each instant in time the number and location of each point of attachment between the virus and the surface and we remove any cleaved sialic acid residues for the remainder of the simulation. Note that we do not prohibit multiple SAs from binding to a single HA centroid; although this would be inaccurate at high densities of SA, under the conditions of these simulations it results in each HA centroid being bound to 0,1, or 2 SAs at any particular instant in time – reasonable values for a trimeric molecule.

## Modeling virus diffusion

To model diffusion of the virus, we use expressions derived for dilute suspensions of cylindrical rods (**Brenner, 1974**):

$$D_{\parallel} = \frac{k_B T \log\left(2L/d - \frac{1}{2}\right)}{2\pi\mu L} \tag{3a}$$

$$D_{\perp} = \frac{k_B T \log\left(2L/d + \frac{1}{2}\right)}{4\pi\mu L} \tag{3b}$$

$$D_{\theta} = \frac{3k_B T \log\left(2L/d - \frac{1}{2}\right)}{\pi\mu L^3} \tag{3c}$$

Where $k_B T$ is the thermal energy scale, $L$ is the virus length, $d$ is the virus diameter, and μ is the solvent viscosity. For a diffusing virus constrained to motion in two dimensions with rotation about its long axis but not its short axis, we determine the change in location and orientation during each time interval, $\Delta t$, using the following expression (**Neild et al., 2010**):

$$\begin{bmatrix} \Delta x \\ \Delta y \\ \Delta\theta \end{bmatrix} = \begin{bmatrix} ccc\left(2D_{\parallel}\Delta t\right)^{\frac{1}{2}}\cos^2\theta + \left(2D_{\perp}\Delta t\right)^{\frac{1}{2}}\sin^2\theta & \left[\left(2D_{\parallel}\Delta t\right)^{\frac{1}{2}} - \left(2D_{\perp}\Delta t\right)^{\frac{1}{2}}\right]\cos\theta\sin\theta & 0 \\ \left[\left(2D_{\parallel}\Delta t\right)^{\frac{1}{2}} - \left(2D_{\perp}\Delta t\right)^{\frac{1}{2}}\right]\cos\theta\sin\theta & \left(2D_{\parallel}\Delta t\right)^{\frac{1}{2}}\sin^2\theta + \left(2D_{\perp}\Delta t\right)^{\frac{1}{2}}\cos^2\theta & 0 \\ 0 & 0 & \left(2D_{\theta}\Delta t\right)^{\frac{1}{2}} \end{bmatrix} \begin{bmatrix} \xi_x \\ \xi_y \\ \xi_\theta \end{bmatrix} \tag{4}$$

Once the virus has formed attachments to SA on the surface, it's translational and rotational motion becomes constrained. To account for this constraint, after calculating the translational and rotational increments during a given time interval, we only accept the resulting values if they do not result in extension of any HA-SA bonds beyond their allowable radius, $b$; otherwise, the simulation time does not advance, and we resample the Langevin parameters {ξx, ξy, ξθ} to take another trial step. As the number of bonds increases, the possible steps that satisfy the constraints of all bonds becomes smaller, resulting in immobilization of the virus. In comparison, in the time it would take a single HA-SA pair to dissociate ($1/k_{off} \sim 1$ s), the typical displacement of an unconstrained virus would be ~1 μm, several orders of magnitude larger than would be tolerated by the constraints of the bond. To reduce the number of times we must resample at each time step, we reduce the translational and rotational diffusion coefficients 100-fold to dampen the virus's motion.

This formulation makes several simplifying approximations, motivated by the cylindrical geometry of viral particles. In particular, we only account for rotations about the virus' major axis, θ, and not around the angles φ or ψ (**Figure 3—figure supplement 4A**). For a cylindrical particle interacting with a flat surface, rotations about the angle ψ would reduce the contact area between the particle and the surface, and would not be favored. This is consistent with our experiments, in which we do not observe filamentous particles pivoting on the surface around the virus's poles. In contrast, rotations around the angle φ would be permissible, although we are unable to detect them experimentally. Because the particles we model are

axisymmetric, rotations about φ do not appreciably change the number or location of HA and NA molecules that are proximal to the surface. As a result, we do not expect that omitting this rotational degree of freedom from our simulations would qualitatively affect the results for cylindrical, axisymmetric particles. However, these simplifying approximations could not be extended to a spherical particle, which could rotate more freely on the surface, altering the number and location of NA molecules capable of cleaving sialic acid.

## Origin of directed motion

In this model, rectification of virus motion comes from asymmetries in the location of HA relative to the SA that are available for binding. At the boundary between regions where NA activity has cleaved SA and regions where it has not, HA molecules will have more potential binding partners in the direction of the uncleaved region. The constraints introduced by binding to these SAs will leave the virus with greater likelihood to step in the direction of the uncleaved region. Although this asymmetrically distributed freedom of motion will only apply to a subset of all HAs, over the course of $\sim 10^6$ simulation steps, it is sufficient to bias the direction of virus diffusion. This establishes a form of Brownian ratchet, where the input energy could derive from the hydrolysis of sialic acid by NA. Any change in NA organization that gets rid of this asymmetry in the orientation of the HA-NA interface (e.g. placing NA at both poles, in a single band away from the poles, spatially separated from HA, or in a uniform distribution over the surface of the virus) abolishes the rectified motion (**Figure 3—figure supplement 5**).

## Effects of sialic acid diffusion

So far, these simulations have used static distributions of SA. Allowing SA to diffuse on the surface will act to dissipate steep gradients in receptor density created by the activity of NA on the virus, resulting in shallower gradients that extend beyond the localized distributions of NA on the viral surface. The extent of this concentration gradient in available receptors will depend on the balance between the catalytic rate of NA, the rate of diffusion of SA, and the size and rate of motion of the virus. These relationships can be parameterized using a non-dimensional Dahmkohler number:

$$Da = \frac{k_{cat}}{D_{SA}/L^2} = \frac{k_{cat}L^2}{D_{SA}} \tag{5}$$

where $k_{cat}$ is the catalytic rate of NA as before ($\sim 100$ s$^{-1}$), $D_{SA}$ is the diffusion coefficient of sialic acid on the surface, and $L$ is a characteristic length for the reaction-diffusion system. Taking $L \sim 100$ nm, $k_{cat} \sim 100$ s$^{-1}$, and $D_{SA} \sim 0.01$–$0.1$ µm$^2$/s (reasonable values for large glycoproteins [**Freeman et al., 2018**; **Jiang et al., 2015**]) gives Da $\sim 10$–$100$. The larger this value is, the better able a virus would be to maintain gradients in sialic acid density to support directional mobility. To test this prediction, we performed simulations where the diffusion coefficient of SA is varied while $k_{cat}$ remains constant and new SA residues are generated at the simulation boundary every time one is cleaved (to prevent complete removal of SA when its diffusion coefficient is high). Consistent with the prediction, simulations of virus motion at Da $\sim 100$ or greater show preferential diffusion along the axis of the virus, while simulations at Da $< 100$ do not (**Figure 3E**). Interestingly, for IAV this transition corresponds to a sialic acid diffusion coefficient of $\sim 0.01$ µm$^2$/s, characteristic of some membrane-anchored glycoproteins (**Jacobson et al., 1984**), but considerably slower than measured values for gangliosides (**Komura et al., 2016**). Although sialic acid diffusion can attenuate persistent virus motion if it is too rapid, it can also increase the rate of motion relative to immobile sialic acid by allowing HAs distal to the HA-NA interface to also sample from asymmetric distributions of receptors (**Figure 3E**). Thus, the nature of the sialic acid receptors to which a virus is bound will likely influence how the virus moves – with greater persistent directional mobility on mucin gels of anchored glycoproteins, and with less persistence once it reaches glycolipids closer to the cell surface.

## Effects of virus length

Increasing the length of the virus increases the number of potential HA-SA interactions, which we would expect to reduce virus mobility if the kinetics of HA-SA binding are held constant. This trend is supported by simulations; varying particle length (185 nm, 285 nm, or 385 nm) while keeping HA-SA binding kinetics constant dramatically reduces particle mobility (**Figure 3—figure supplement 6**, top row), whereas holding the average number of HA-SA bonds constant results in slightly higher persistent directional mobility for larger particles (**Figure 3—figure supplement 6**, bottom row). In both cases, simulations show a greater tendency for longer viruses to maintain constant orientation. Thus, although particle size and HA-SA binding kinetics both alter virus motion quantitatively, the qualitative features of directional mobility described throughout this work are preserved in particles approaching the lower end of the length distributions observed for filamentous IAV.

## First-passage model of virus transport through mucus

The diffusion of viral particles through mucus can be modeled as a one-dimensional problem, where particles initially binds to the distal part of a mucus gel with thickness Δ and diffuse until reaching the surface of the underlying epithelial cells. Solving the diffusion equation with reflecting boundary conditions at $x = \Delta$ (modeling the tendency of viruses to stay bound to the mucus layer) and absorbing boundary conditions at $x = 0$ (modeling transfer of the viral particle from the mucus gel to the cell surface) gives the probability density of particles within the mucus gel as a function of time:

$$p(x,t) = \sum_n 2\sin\left(\frac{n\pi}{2}\right)exp\left(-\frac{n^2\pi^2 D_{virus}}{4\Delta^2}t\right)\sin\left(\frac{n\pi}{2\Delta}x\right) \tag{6}$$

Integrating this expression over $x$ gives the probability that a virus will remain in the mucus gel at time $t$, and subtracting this quantity from one gives the probability, $P(t)$, that a virus will have reached the cell surface by time $t$:

$$P(t) = 1 - \sum_n \frac{4\sin\left(\frac{n\pi}{2}\right)}{n\pi}exp\left(-\frac{n^2\pi^2 D_{virus}}{4\Delta^2}t\right) \tag{7}$$

This expression allows us to estimate the survival probability of a virus in the presence of mucociliary clearance by evaluating $P(t = L/U)$, where $L$ (~10 cm) is the distance a bound particle must be carried to be neutralized and $U$ (~100 μm/s) is the velocity of mucociliary transport. This will depend on the virus diffusion coefficient, $D_{virus}$, which we estimate from the data in **Figure 3** and the relationship $D_{virus}$ = MSD/4 t. This gives a value of ~ 800 nm$^2$/s in two dimensions or ~ 400 nm$^2$/s in one dimension, which we can substitute into the expressions above. The results of these calculations are plotted in **Figure 4—figure supplement 1**, where we compare the first passage probability of viruses with five-fold differences in their diffusion coefficients in a mucus layer that is 2 μm thick. For the parameters listed here, the timescale for mucociliary clearance (~20 min) is substantially shorter than the timescale for virus diffusion (~2 hr), and the percentage of successful particles (defined for simplicity as those that reach the epithelial surface) is extremely sensitive to their diffusion coefficient: the five-fold decrease in $D_{virus}$ that we estimate would result from the loss of persistent motion would lead to > 1000 fold decrease in the number of successful particles (**Figure 4—figure supplement 1B**). This calculation assumes that binding of viruses to the cell surface is immediate and irreversible – assumptions that are reasonable if the cell surface is densely decorated with sialic acid and if viruses that come in contact with it are rapidly endocytosed. However, regardless of how interactions between viruses and the underlying epithelium are modeled, the rate at which viruses encounter the epithelium – our primary focus, and a prerequisite for productive infection - will always be inversely proportional to the virus's diffusion coefficient. Although this simple model provides only a rough estimate, this analysis, which is based on reasonable estimates of physiological parameters (**Bustamante-Marin and Ostrowski, 2017**), suggests that persistent motion could enhance the frequency of host-to-host transmission by orders of magnitude.

