## [Decision Letter]

Thank you for submitting your article "Influenza A virus surface proteins are organized to help penetrate host mucus" for consideration by *eLife*. Your article has been reviewed by three peer reviewers, including Richard A Neher as the Reviewing Editor and Reviewer #1, and the evaluation has been overseen by Arup Chakraborty as the Senior Editor. The following individual involved in review of your submission has agreed to reveal their identity: Mark Zanin (Reviewer #2).

The reviewers have discussed the reviews with one another and the Reviewing Editor has drafted this decision to help you prepare a revised submission.

Summary:

In this manuscript, Vahey and Fletcher investigate the importance of the spatial organization of hemagglutinin and neuraminidase in facilitating the movement of influenza A viruses (IAVs) through mucus. They report that neuraminidase proteins cluster at one pole of the virus and this clustering asymmetrically frees the virion from tethering to sialic acid, thereby facilitating directed motion. This is an interesting novel finding that provides insights into outstanding questions in the field. However, a number of issues need to be addressed before we can recommend publication.

Essential revisions:

The issues raised during review and the ensuing discussion broadly fall into four categories concerning i) further quantification, ii) the theoretical model, iii) direct assays of infectivity, and (iv) availability of analysis code.

1) A number of statements would benefit from more quantitative analysis.

a) Figure 1 should show distributions of data rather than representative examples. Figure 1A could, for example, be backed up by two histograms showing NA/HA intensities in the first/last 20% of the virion. Or last/first 100nm or similar. Figure 1D needs a distribution of correlation coefficients of HA/NA. Showing one example is simply not enough. Figure 1E should show distributions as well. Bar graphs are not good.

b) Figure 3A-C: While the measurement of the persistence of orientation is useful and interesting, we would like to see a quantitative and direct assessment of diffusivity, for example by comparing MSD vs time for the deletion mutant and the WT NA. It should be quantified whether the morphology of the viruses change when the cytoplasmic tail of NA is deleted. If so, does this affect the ability to determine the orientation of the particles?

c) Figure 4C D: Again, you should try to find a meaningful quantitative comparison rather than just showing examples.

d) Other statements need more precision: E.g. Results section, paragraph one: How were polarized viruses selected? What fraction of particles were filamentous? etc.

2) The theoretical model should be explained more clearly and connected better to the experimental results.

a) We suggest moving some of the descriptions of the model into the main text and maybe include an illustration of the model, the sialic acid distribution along the particle, and the resulting directional motion. It would also help to discuss in more simple and explicit terms that how macroscopic diffusion increases with more directionally persistent microscopic motion.

b) When describing the different physical effects, you should be careful not to confuse the reader with diffusion constants of NA, receptors, and viruses all labeled D. Please differentiate them with different symbols or subscripts.

c) The model prediction that very slow sialic acid diffusion results in slow virus diffusion should be better explained. How does this prediction compare with the experiments in Figure 2 using coverslips functionalized with biotin-anchored (and presumably immobile) receptors?

d) Please point out that Figure 4—figure supplement 1 essentially only shows the first-passage probability calculated in the supplement (please number equations and refer to them directly). This figure could be improved by changing the x-scale seconds -> minutes and illustrating more generally how increasing virus diffusion increases the probability of reaching the epithelium. You could add several curves for different D, plot this on a log scale, and remove the 5000x.

e) The assumption of an absorbing boundary at the epithelium implies that binding is essentially instantaneous and irreversible. This should be discussed.

3) Additional infectivity and/or mobility assays.

a) Could infectivity be assessed directly using mucus-producing cells such as CaLu-3? Such data would considerably strengthen the claim that enhanced diffusion increased infectivity.

b) Enhanced diffusion in polarized filamentous virus compared to spherical viruses is counteracted by their larger size. Is there a head-to-head comparison?

c) The data in this paper was obtained using viruses containing fluorescent labels. Have the authors confirmed these observations using unlabeled viruses? Whilst obviously fluorescent microscopy could not be conducted with unlabeled viruses, the morphology of virions propagated in mucus-producing cells could be compared to those propagated in non-mucus producing cells and studies of their ECL tracks could possibly be conducted. This experiment could add further weight to their statement that filamentous morphologies are adaptations to replication in the presence of mucus.

4) Simulation code and analysis scripts need to be made available, preferably on a repository like GitHub with appropriate documentation.

[Editors' note: further revisions were requested prior to acceptance, as described below.]

Thank you for resubmitting your work entitled "Influenza A virus surface proteins are organized to help penetrate host mucus" for further consideration at *eLife*. Your revised article has been favorably evaluated by Arup Chakraborty (Senior Editor) and a Reviewing Editor.

By and large, the authors have presented a compelling revision. They have quantified a number of previously rather qualitative statements by showing distributions rather than representative examples, present additional data on virus mobility directly comparing NA and NA-DCT viruses, streamlined notation, and explained the theoretical model better. However, there are a number of remaining issues that we would like to have clarified/rectified.

Figure 1: the distributions are useful, but the alignment of the NA rich pole with NP seems rather imprecise. Reading the previous version of the manuscript, I was certainly expecting a more clear cut picture. The insets in Figure 1A also suggest a much more pronounced asymmetry. How closely is this more blurry picture accounted for in the simulation?

Figure 1E, Anticorrelation of HA and NA: This anticorrelation is pretty weak and the main take home message from the figure seems to be that HA and NA are clustered. Hence I think the statement "...NA clusters that appear to largely exclude HA" is too strong.

Figure 1G seems problematic. The caption states that p-values are calculated using a two-sample t-test. What enters as independent data point here? Strictly speaking, you have n=4 replicates and I doubt that this would support the conclusion that these cases are different. A paired test on +/- NAI samples from the same replicate would probably be more powerful. Generally, quantifying the difference (and confidence intervals) between two conditions is preferable to rejecting a null.

The addition to Figure 4E does not help. This figure shows that there is little evidence for variation of ECL intensity and HA-NA polarity. Making an arbitrary cut at 0.06 and fitting lines to points below and above is not appropriate. Furthermore, a correlation of ECL intensity and HA-NA polarity doesn't quantify the examples given in C or D. You could try to show ECL intensity distributions aligned with the NA polarity or similar. But as of now, panels C, D, and E are just examples.

The comparisons of diffusion with/without correlations make sense only when you specify the time interval over which the displacements are measured (paragraph three of subsection “Polarized viruses step persistently away from their NA-rich pole”). You do so later, but I'd suggest moving it forward.

---

## [Author Response]

Essential revisions:The issues raised during review and the ensuing discussion broadly fall into four categories concerning i) further quantification, ii) the theoretical model, iii) direct assays of infectivity, and (iv) availability of analysis code.1) A number of statements would benefit from more quantitative analysis.a) Figure 1 should show distributions of data rather than representative examples. Figure 1A could, for example, be backed up by two histograms showing NA/HA intensities in the first/last 20% of the virion. Or last/first 100nm or similar. Figure 1D needs a distribution of correlation coefficients of HA/NA. Showing one example is simply not enough. Figure 1E should show distributions as well. Bar graphs are not good.

We have revised Figure 1 to further quantify our results, and to display data as distributions rather than isolated examples or bar graphs. Specifically, in Figure 1A we have added histograms showing relative HA and NA intensities in the outermost ~400nm of each virus particle (since virus lengths vary considerably, we believe that analyzing intensities within a fixed distance of the viral poles is preferable to analyzing within a fixed percentage of virus length). We have also performed this analysis for the subset of the population that contain NP at one end of the particle, by aligning each particle to the NP-containing pole. This analysis (shown in Figure 1A, right) replaces the composite image of a virus population that was previously presented in Figure 1B and which is now included as Figure 1—figure supplement 1.

Additionally, we have added a new panel, Figure 1E, which shows correlation coefficients for spatial distributions of HA and NA from STORM images of 24 viruses. This additional analysis supports the observation that HA and NA are spatially segregated on the surface of the virus. Finally, we have revised our presentation of data from Figure 1E (now Figure 1G in the revised manuscript) to show distributions of data from each of the four biological replicates. We have revised the initial paragraphs of the Results section to reflect these additions, and to streamline and clarify our description of the data.

b) Figure 3A-C: While the measurement of the persistence of orientation is useful and interesting, we would like to see a quantitative and direct assessment of diffusivity, for example by comparing MSD vs time for the deletion mutant and the WT NA. It should be quantified whether the morphology of the viruses change when the cytoplasmic tail of NA is deleted. If so, does this affect the ability to determine the orientation of the particles?

We agree with the reviewer’s suggestion than quantitative comparisons of diffusion coefficients are critical to understand the consequences of deleting the NA cytoplasmic tail. To facilitate this comparison, we have moved the plots showing mean squared displacement versus time for the wildtype virus from Figure 2D (in the previous manuscript) to Figure 3B (in the revised manuscript), where they are now plotted alongside the corresponding data for the cytoplasmic tail deletion mutant. These results show that wildtype virus has a higher diffusion coefficient than NAΔCT virus, but that this situation is reversed when directional correlations are computationally removed from the data. This result highlights the importance of directional persistence in enhancing the diffusion of wildtype viruses.

Separately, we have also measured the size distribution of viruses with wildtype NA or ΔCT NA. This data, which is presented in Figure 3—figure supplement 2C, shows that the length distributions are similar up to a length of ~3μm, beyond which the wildtype virus is more highly represented. This result is also conveyed qualitatively in Figure 3—figure supplement 1C, which shows images of the two types of virus. Because both types of virus produce filamentous particles with well-defined major and minor axes, this difference in the abundance of extremely large particles (which represent a minor proportion of our data) does not affect our ability to determine particle orientation. To clarify that wildtype and NAΔCT are similar – but not identical - in morphology, we have revised the text to include the statement “…both wildtype and NAΔCT viruses form filamentous particles (with wildtype virus producing a higher proportion of extremely large filaments, >3μm in length; Figure 3—figure supplement 2C).”

c) Figure 4C and D: Again, you should try to find a meaningful quantitative comparison rather than just showing examples.

To quantify ECL tracks left in mucus more comprehensively, we measured the HA-NA polarity of viral particles and the intensity of ECL labeling associated with each particle. The results of this analysis are now plotted as Figure 4E, which shows the relationship between HA-NA polarity, and the extent of ECL labeling in the surrounding mucus. Referring to this data in the text, we write: “Overall, we find that the amount of cleaved sialic acid in proximity to a viral particle tends to increase with HA-NA polarity beyond a threshold value of ~0.06 (Figure 4E).”

d) Other statements need more precision: E.g. Results section, paragraph one: How were polarized viruses selected? What fraction of particles were filamentous? etc.

We have revised the manuscript to make statements categorizing virus particles more quantitatively precise.

Regarding the virus used, we write: “For these experiments, we use a tagged variant of the strain A/WSN/1933 with M1 from A/Udorn/1972, which differs from WSN M1 at six residues and confers filamentous morphology (Elleman and Barclay, 2004). Consistent with our prior observations, this virus produces filamentous particles that vary widely in size, from sub-diffraction limited spots to particles >10μm in length (Vahey and Fletcher, 2019).” To address the fraction of filamentous particles, we have included a plot of particle length distributions in Figure 3—figure supplement 2C. By describing particle morphology in this way (rather than reporting the fraction of particles that are filamentous), we avoid the need to establish an arbitrary cutoff between filamentous and non-filamentous morphology.

To make the distinction between mobile and non-mobile particles more quantitative and precise, we have revised our analysis to define these categories according to measured diffusion coefficients. Specifically, we first calculate a diffusion coefficient for each particle from its squared displacement versus time, as measured from time lapse videos. This results in a bimodal distribution of measured diffusion coefficients, which we plot in Figure 2C in the revised manuscript. Based on this distribution, we define mobile particles as those with diffusion coefficients >100nm^2^/s, corresponding to ~68% of the particles measured. The remaining particles are categorized as non-mobile. We use the same criteria in our analysis of NAΔCT virus in Figure 3B and C, taking the mobile population as those with diffusion coefficients >100nm^2^/s.

Regarding the reviewers’ question of how polarized viruses were selected, we note that, while HA-NA polarity is quantified in several figures (Figure 1G, 2E, 4E), we did not use this as a metric for selecting particles in any of our experiments. Rather, we characterized and reported the polarity of the viruses in the populations of each experiment.

2) The theoretical model should be explained more clearly and connected better to the experimental results.a) We suggest moving some of the descriptions of the model into the main text and maybe include an illustration of the model, the sialic acid distribution along the particle, and the resulting directional motion. It would also help to discuss in more simple and explicit terms that how macroscopic diffusion increases with more directionally persistent microscopic motion.

To explain the theoretical model more clearly and intuitively, we have revised its discussion in the main text and provided an additional supplementary figure (Figure 3—figure supplement 5) which illustrates the distribution of cleaved sialic acid that is created by bound viruses that have different distributions of NA on their surface.

To provide a simple and intuitive analogy for how macroscopic diffusion increases with more directionally persistent microscopic motion, we have revised the text to include an analogy between our experiments and the persistence length of a polymer chain. Specifically, in the paragraph comparing the MSDs of wildtype and ΔCT virus, we have added the sentence: “Correlations in the direction of successive steps in a random walk will increase the rate at which the walker explores its environment, analogous to how more rigid polymer chains typically have larger end-to-end distances than those with equivalent contour length but lower bending rigidity.”

Finally, we appreciate the reviewer’s suggestion to move additional material from the supplement. However, we were unsatisfied by our attempts to combine this material with the main text of the manuscript and feel that it is best left as an appendix. We hope that in organizing the manuscript in this way we have made our key results easier to follow, while still providing a detailed description of all aspects of the work.

b) When describing the different physical effects, you should be careful not to confuse the reader with diffusion constants of NA, receptors, and viruses all labeled D. Please differentiate them with different symbols or subscripts.

We appreciate the possible confusion regarding the use of D to define the multiple distinct diffusion coefficients. We have revised the manuscript text to refer to sialic acid diffusion coefficients as DSA, and virus diffusion coefficients as *D_virus_*, and we have additionally attempted to make it clear from context what we are referring to in each case.

c) The model prediction that very slow sialic acid diffusion results in slow virus diffusion should be better explained. How does this prediction compare with the experiments in Figure 2 using coverslips functionalized with biotin-anchored (and presumably immobile) receptors?

As the reviewers correctly point out, our simulations predict that virus diffusion is reduced when sialic acid is immobilized, a scenario that models our experimental results using biotin-anchored fetuin. The origin for the enhancement in virus mobility that we predict when the sialic acid diffusion coefficient is low but non-zero is the ability of HAs along a greater portion of the virus to sample from spatially non-uniform distributions of sialic acid. In contrast, when the sialic acid is immobilized, only those HAs adjacent to the NA-rich pole sample from non-uniform distributions of binding partners.

We have added text to clarify this point. In the main article, we write: “Although our experimental data corresponds to the case where sialic acid is immobilized (DSA ~ 0), our simulations predict that IAV bound to slowly-diffusing transmembrane glycoproteins gel-forming mucins (DSA <0.01µm^2^/s) will exhibit enhanced directed mobility, likely due to the biased spatial distributions of sialic acid binding partners that are available to HAs away from the virus’s NA-rich pole.” This point is reinforced in the text, where we describe this result: “Although sialic acid diffusion can attenuate persistent virus motion if it is too rapid, it can also increase the rate of motion relative to immobile sialic acid by allowing HAs distal to the HA-NA interface to also sample from asymmetric distributions of receptors (Figure 3E).”

d) Please point out that Figure 4—figure supplement 1 essentially only shows the first-passage probability calculated in the supplement (please number equations and refer to them directly). This figure could be improved by changing the x-scale seconds -> minutes and illustrating more generally how increasing virus diffusion increases the probability of reaching the epithelium. You could add several curves for different D, plot this on a log scale, and remove the 5000x.

We have revised Figure 4—figure supplement 1 as suggested by the reviewers, changing the x-axis from seconds to minutes, changing the vertical axis to log scale, and plotting the solution for a range of virus diffusion coefficients (80nm^2^/s to 800nm^2^/s at intervals of 80nm^2^/s). We have also numbered all equations in the manuscript.

We have also attempted to make it clear that these plots show the results of the first-passage model discussed in the text; in both the original and revised manuscript, we note this in the sentence where we refer to the figure: “…adapting this value for a one-dimensional first passage model suggests that persistent motion increases the number of particles that reach the epithelium before being cleared by several orders of magnitude (Figure 4—figure supplement 1B).” This is also mentioned in the text: “The results of these calculations are plotted in Figure 4—figure supplement 1, where we compare the first passage probability of viruses with five-fold differences in their diffusion coefficients in a mucus layer that is 2 μm thick.” Finally, the title of the figure - “a first-passage model for virus transport in mucus” – also states that the results displayed correspond to a first-passage probability, and we have added this label directly to the axis of the plot in Figure 4—figure supplement 1B.

e) The assumption of an absorbing boundary at the epithelium implies that binding is essentially instantaneous and irreversible. This should be discussed.

The reviewers correctly point out an implicit assumption we made in using our first-passage model to estimate the percentage of “successful” virus particles – that the epithelium acts as an absorbing boundary. We have added the following discussion of this point to the text:

“For the parameters listed here, the timescale for mucociliary clearance (~20 min) is substantially shorter than the timescale for virus diffusion (~2h), and the percentage of successful particles (defined for simplicity as those that reach the epithelial surface) is extremely sensitive to their diffusion coefficient: the five-fold decrease in *D_virus_* that we estimate would result from the loss of persistent motion would lead to >1000-fold decrease in the number of successful particles (Figure 4—figure supplement 1B). This calculation assumes that binding of viruses to the cell surface is immediate and irreversible – assumptions that are reasonable if the cell surface is densely decorated with sialic acid and if viruses that come in contact with it are rapidly endocytosed. However, regardless of how interactions between viruses and the underlying epithelium are modeled, the rate at which viruses encounter the epithelium – our primary focus, and a prerequisite for productive infection - will always be inversely proportional to the virus’s diffusion coefficient.”

3) Additional infectivity and/or mobility assays.a) Could infectivity be assessed directly using mucus-producing cells such as CaLu-3? Such data would considerably strengthen the claim that enhanced diffusion increased infectivity.

As suggested by the reviewers, one hypothesis suggested by this work is that virus particles that exhibit polarized distributions of HA and NA (and thus a tendency towards directionally-persistent diffusion) may have higher infectivity in mucosal environments. Testing this hypothesis definitively would require measuring the infectious potential of virus particles in mucus as a function of their shape and molecular organization, ideally with other features of the virus held constant. One way to accomplish this could be to sort virus particles from the same genetic background into polarized and non-polarized subsets, and to measure the particle-to-PFU ratio for each as a measure of infectious potential. Alternatively, another possibility would be to visualize infection of mucus-producing cells directly, allowing us to measure rates of infection for different virus phenotypes relative to their frequency within the overall population.

Although we agree that these experiments would be informative and worthwhile, they present technical challenges and are currently outside our experimental capabilities. Because of this limitation, our intention in both the original and revised manuscript has been to limit our claims to only the characteristics we are able to measure (i.e. diffusional characteristics of virus particles), without making the claim that these characteristics enhance infectivity. To clarify this point, we have revised the manuscript to use more neutral language when describing persistent diffusion, and to be more explicit about what our results do not allow us to conclude:

First paragraph of Discussion: “this advantage of filamentous morphology" revised to “this feature of filamentous morphology".

Second paragraph of Discussion: “Although more work is needed to understand if these characteristics of IAV organization are adaptive during in vivo replication” revised to “Although more work is needed to understand if these characteristics of IAV organization enhance infectivity and are adaptive during in vivo replication”.

b) Enhanced diffusion in polarized filamentous virus compared to spherical viruses is counteracted by their larger size. Is there a head-to-head comparison?

The reviewers correctly point out that, with everything else held constant, particle mobility will tend to decrease with increasing particle size. Although filamentous particles can be much larger than spherical ones (and would thus be expected to have lower mobility), this is not universally the case; filamentous strains of IAV produce pill-shaped particles as small as 120nm in length, with a diameter of ~80nm measured from the outermost glycoprotein layer (Calder et al., 2010). Although these viruses are smaller in terms of surface area than many spherical influenza particles, their lack of radial symmetry is shared by other filamentous particles and suggests that they could still support polarized distributions of HA and NA and exhibit the directional diffusion observed here in larger particles. However, this is challenging to confirm experimentally, as we are only able to characterize the features of particles with dimensions below the diffraction limit of our microscope (~300nm) using super-resolution methods that require fixation.

Although we do not have direct head-to-head comparison of spherical and filamentous IAV, we explored how particle size influences mobility computationally in Figure 3—figure supplement 6 of the revised manuscript (Figure 3—figure supplement 5 in the original submission). Here, our simulations show that particle mobility decreases for larger particles when the HA-sialic acid binding affinity is held constant, but that particle mobility increases for filamentous particle if instead the number of HA-sialic acid bonds is held constant. Similar simulations of spherical particles would also be interesting; however, as discussed in Figure 3—figure supplement 4 in the original and revised manuscript, the dynamics of spherical particles on two-dimensional surfaces would need to account for particle rolling, which we have neglected for simplicity.

We have revised the manuscript to clarify this point and the limits of our experimental results. Specifically, we have revised the first paragraph in the Discussion to read in part:

“Importantly, our simulations show that this feature of filamentous morphology would not be limited to the extremely large particles whose length is easily measured using diffraction limited microscopy, but rather extends to capsule-shaped particles <200nm in length (Figure 3—figure supplement 5), which are known to be produced by filamentous strains of IAV (Calder et al., 2010). In contrast, this effect may be suppressed in spherical particles, due both to the absence of a clear axis for polarization and their ability to roll on two-dimensional surfaces. However, more work is needed using higher-resolution live imaging to confirm these predictions experimentally.”

c) The data in this paper was obtained using viruses containing fluorescent labels. Have the authors confirmed these observations using unlabeled viruses? Whilst obviously fluorescent microscopy could not be conducted with unlabeled viruses, the morphology of virions propagated in mucus-producing cells could be compared to those propagated in non-mucus producing cells and studies of their ECL tracks could possibly be conducted. This experiment could add further weight to their statement that filamentous morphologies are adaptations to replication in the presence of mucus.

The reviewers here raise two important points. The first point regards the effect of labeling on virus mobility. We have sought to address this comment in three complementary ways. In our prior work, we measured the effect of virus labeling on infectivity, finding that virus with fluorophores on both HA and NA retain ~85% infectivity relative to unlabeled virus (Figure 1F from Vahey et al., 2019). The observation that labeled viruses bind to and enter cells with efficiency similar to unmodified viruses suggests that the labeling is not substantially perturbing the virus’s ability to engage with sialic acid at physiological densities on the cell surface. Additionally, in the present work we have characterized the effect of labeling on NA activity, using the fluorogenic substrate MUNANA (Figure 2—figure supplement 1 in both the original and revised manuscript). These results demonstrate that NA activity is not altered by the attachment of Alexafluor 555, the fluorescent dye used for functional assays throughout this work. Finally, ECL labeling of tracks left by unlabeled virus resemble those left by labeled virus (Author response image 1). Collectively, these experiments give us confidence that the labeling strategy employed throughout this work does not influence our results.

**Author response image 1. respfig1:** ECL tracks from unlabeled virus. Filamentous viruses unmodified by fluorophores leave persistent trails of cleaved sialic acid (detected by labeling with ECL and displayed here with inverted contrast) similar to those left by fluorescently labeled viruses (Fig. 2B in the revised manuscript).

The second point the reviewers raise here regards the characteristics of virus raised in the presence or absence of mucus; specifically, whether we could compare the morphology and diffusional characteristics of two different virus populations grown under these two conditions. These experiments would provide new insight into the environmental pressures that contribute to virus morphology, and they could help to test the hypothesis discussed in point 3a of this response - that filamentous morphology combined with polarized distributions of HA and NA is adaptive during replication in the presence of mucus. Although we agree that this would be a valuable contribution, we feel that these experiments are outside the scope of this work, which focuses more narrowly on mechanisms of virus diffusion.

4) Simulation code and analysis scripts need to be made available, preferably on a repository like GitHub with appropriate documentation.

We have compiled and documented Matlab data and code for generating plots from the manuscript, and for running simulations of virus diffusion. These are found in five compressed supplemental folders: Figure 1—source data 1, Figure 2—source data 1, Figure 3—source data 1, Figure 4—source data 1, and Supplementary code 1.

[Editors' note: further revisions were requested prior to acceptance, as described below.]

By and large, the authors have presented a compelling revision. They have quantified a number of previously rather qualitative statements by showing distributions rather than representative examples, present additional data on virus mobility directly comparing NA and NA-DCT viruses, streamlined notation, and explained the theoretical model better. However, there are a number of remaining issues that we would like to have clarified/rectified.

We thank the reviewers for their favorable response, and for their continued evaluation of our manuscript. Our efforts to address the comments on our revision are described below and highlighted in the revised manuscript.

Figure 1: the distributions are useful, but the alignment of the NA rich pole with NP seems rather imprecise. Reading the previous version of the manuscript, I was certainly expecting a more clear cut picture. The insets in Figure 1A also suggest a much more pronounced asymmetry. How closely is this more blurry picture accounted for in the simulation?

As the reviewer’s point out, the virus population is quite heterogeneous in a number of regards, including the polarity of the NA distribution and its alignment with NP. As we report in the manuscript, NA is enriched >5-fold at one pole relative to the other pole in ~34% of particles, and NA is more abundant at the NP-containing pole in ~70% of cases. We believe that this variability is a natural (and interesting) feature of influenza A virus assembly.

To better calibrate the reader to the examples shown in the inset images in Figure 1A, we have added values for the NA enrichment at one pole relative to the other for both cases (this information is in the caption of the most recent revision). For the inset to the left, NA is enriched 5.6-fold at one pole relative to the other, while in the inset to the right, the corresponding enrichment is 3.3-fold. These values are within the 31st and 48th percentile of the total population, respectively. Thus, while each of these cases show a pronounced asymmetry (as remarked by the reviewer), they fall reasonably within the observed distributions. Indeed, nearly half of the virus population has a more polarized distribution of NA than the example shown in the inset to the right.

Of the particles with less polarized distributions of NA (i.e. a ratio of NA abundance at the two poles of ~1), some have more uniform distributions of NA along the entire particle length while others have bipolar distributions, with NA enriched at both poles. For both of these cases, we have performed simulations (Figure 3—figure supplement 5D) that show that uniform or bipolar particles do not exhibit directional mobility. To highlight this connection between these simulation results and heterogeneity we observe in the virus population, we have revised our discussion of Figure 3—figure supplement 5D to read:

“Consistent with this mechanism, persistent motion of the virus is lost when NA is no longer localized to a single virus pole and the sialic acid distribution beneath the particle becomes symmetric (Figure 3—figure supplement 5B-D). We note that virions with symmetric distributions of NA also occur in the virus population and would not be expected to exhibit persistent motion (for example, those particles with NA polar ratios of ~1 in Figure 1A).”

Figure 1E, Anticorrelation of HA and NA: This anticorrelation is pretty weak and the main take home message from the figure seems to be that HA and NA are clustered. Hence I think the statement "...NA clusters that appear to largely exclude HA" is too strong.

We agree that our results more clearly show that HA and NA are clustered on the virus surface, and that conclusively demonstrating exclusion of HA from NA clusters (or vice versa) would be challenging without further improvements in spatial resolution. Accordingly, we have revised the text to read:

“We also find that the NA seen at low levels along the length of filamentous viruses without super-resolution imaging is actually organized into small NA clusters (Figure 1D). Additionally, HA and NA distributions in these particles are modestly anti-correlated (Figure 1E), suggesting that receptor-binding and receptor-destroying activities may be spatially segregated on some regions of the virus.”

At least two considerations could lead the correlation coefficients that we measure to be conservative estimates and contribute to what the reviewer notes is a “pretty weak” anticorrelation. First, the finite resolution achievable using fluorescence microscopy (both conventional and super-resolution) will tend to obscure differences in spatial distributions of molecules that are densely packed and closely apposed. This optical blurring will necessarily make spatial correlations less negative. Second, our STORM reconstructions are two-dimensional projections of a three-dimensional object. Thus, if the virus is not axially symmetric in its protein distributions, the spatial correlations will become less negative due to spurious colocalization of molecules that are actually on opposite sides of the particle.

Figure 1G seems problematic. The caption states that p-values are calculated using a two-sample t-test. What enters as independent data point here? Strictly speaking, you have n=4 replicates and I doubt that this would support the conclusion that these cases are different. A paired test on +/- NAI samples from the same replicate would probably be more powerful. Generally, quantifying the difference (and confidence intervals) between two conditions is preferable to rejecting a null.

We appreciate the reviewer’s feedback on the presentation and statistical analysis for Figure 1G. In our analysis for this figure, the median value of each biological replicate for a given condition represents an independent data point. In our previous submissions, the p-values displayed in the figure were determined from a two-sample t-test based on these data points (four per condition).

In the revised manuscript, we have updated Figure 1G to show the results of a paired t-test, and we have added confidence intervals to our discussion of this data in the main text. Both the paired t-test and the two-sample t-test support the claim that viruses that detach from the cell surface during NAI treatment are more polarized than those that remain sequestered on the cell surface. We have provided the source data for this figure so that further quantitative analysis of our data can be carried out by anyone interested.

The addition to Figure 4E does not help. This figure shows that there is little evidence for variation of ECL intensity and HA-NA polarity. Making an arbitrary cut at 0.06 and fitting lines to points below and above is not appropriate. Furthermore, a correlation of ECL intensity and HA-NA polarity doesn't quantify the examples given in C or D. You could try to show ECL intensity distributions aligned with the NA polarity or similar. But as of now, panels C, D, and E are just examples.

Based on the reviewer’s suggestion, we have re-analyzed data for Figure 4 to examine alignment of the virus’s HA-NA axis with the axis of ECL labeling (defined as the vector connecting the virus center to the center of ECL labeling). As plotted in Figure 4E of the current revision, we find that the number of particles aligned within 60^o^ of parallel to the ECL axis are about 1.5-fold more abundant than the number of particles aligned within 60^o^ of antiparallel. This result is qualitatively similar to the plots shown in Figure 3C and D.

While the relationships we measure in this figure are modest, it is worth considering technical aspects of the native mucus gel experiments that pose limits on our analysis. Unlike the time-resolved, two-dimensional particle trajectories of the idealized reconstitution experiments that serve as the basis for analysis in figures 2 and 3, the experiments in Figure 4 only provide the position and bearing of a virus at one point in time: the instant at which we fix the sample. In addition, viruses in these experiments are free to move in three dimensions and may sit relatively far from the coverslip, making the samples more challenging to image and quantify. Finally, we expect that the mucus itself will diffuse and flow during the experiment, altering distributions of cleaved sialic acid in ways that we are unable to predict or measure. Despite these experimental caveats, we believe that the results presented in this figure provide valuable physiological context to the more quantitative measurements described elsewhere in the manuscript. To avoid confusion about what these results do and do not show, we have revised our description of Figure 4C-E in the main text, writing:

“This suggests that the asymmetry of NA and HA biases the direction of virus diffusion in three dimensions, producing trails of cleaved sialic acid that can reach several microns in length as the virus moves (Figure 4C). This directional mobility is less prevalent in viruses with more uniformly distributed NA, though they nonetheless are capable of creating swaths of ECL-staining within the mucus that lack clear directionality (Figure 4D). Similar to our observations on idealized two-dimensional surfaces (Figure 3C), we find that viruses in mucus exhibit a slight tendency to align their HA-NA axis (captured at the moment of fixation) with the displacement of the virus relative to the center of ECL labeling (Figure 4E). These results suggest that the spatial organization of HA and NA on the virus surface may also promote penetration of polarized IAV particles through mucus barriers in vivo.”

Lastly, we agree with the reviewer that Figure 4C and D are just examples, and we have clarified our discussion of these panels in the text and caption to make this caveat clear. To place these examples in the context of Figure 4E (where we plot data from the entire population), we have added the angular alignment for each particle relative to its ECL trail in the lower left corner of each panel. Because the example images in Figure 4D contain multiple particles in contact with overlapping ECL trails, we have not reported angular alignments in this case; as described in the Materials and methods section, these cases are also excluded from the population analysis in Figure 4E. The data for this plot is also included as a Source Data file for further analysis by interested individuals.

The comparisons of diffusion with/without correlations make sense only when you specify the time interval over which the displacements are measured (paragraph three of subsection “Polarized viruses step persistently away from their NA-rich pole”). You do so later, but I'd suggest moving it forward.

We have revised this section of the manuscript, writing that we “simulated random walks in which the number of steps, the size of each step, and the time interval between steps all match the observed data, but the direction of each step is uncorrelated with previous steps (Figure 3B).”